# Navigating the Maze of Explainable AI:
# A Systematic Approach to Evaluating Methods and Metrics

## Abstract

Explainable AI (XAI) is a rapidly growing domain with a myriad of proposed methods as well as metrics aiming to evaluate their efficacy. However, current literature is often of limited scope, examining only a handful of XAI methods and ignoring underlying design parameters for performance, such as the model architecture or the nature of input data. Moreover, they often rely on one or a few metrics and neglect thorough validation, increasing the risk of selection bias and ignoring discrepancies among metrics. These shortcomings leave practitioners confused about which method to choose for their problem. In response, we introduce LATEC, a large-scale benchmark that critically evaluates 17 prominent XAI methods using 20 distinct metrics. We systematically incorporate vital design parameters like varied architectures and diverse input modalities, resulting in 7,560 examined combinations. Through LATEC, we showcase the high risk of conflicting metrics leading to unreliable rankings and consequently propose a more robust evaluation procedure. Further, we comprehensively evaluate various XAI methods to assist practitioners in selecting appropriate methods aligning with their needs. Curiously, the emerging top-performing method, Expected Gradients, is not examined in any relevant related study. LATEC reinforces its role in future XAI research by publicly releasing all auxiliary data, including model weights, over 326k saliency maps, and 378k metric scores as a dataset. The benchmark is hosted at: `https://github.com/kjdhfg/LATEC`.

## 1 Introduction

Explainable AI (XAI) methods have become essential tools in numerous domains, allowing for a better understanding of complex machine learning decisions. The most prevalent XAI methods originate from the domain of saliency maps (Simonyan et al., 2013). As the diversity and abundance of proposed saliency XAI methods expand alongside their growing popularity, ensuring their reliability becomes paramount (Adebayo et al., 2018). Given that there is no clear "ground truth" for individual explanations (e.g., discussed in Adebayo et al. (2020)), the trustworthiness of XAI methods is typically determined by examining three key criteria: their accuracy in reflecting a model's reasoning ("faithfulness") (Bach et al., 2015; Samek et al., 2017), their stability under small changes ("robustness") (Yeh et al., 2019; Alvarez Melis & Jaakkola, 2018), and the understandability of their explanations ("complexity") (Chalasani et al., 2020; Bhatt et al., 2021). Beyond qualitative assessments such as in Doshi-Velez & Kim (2017); Ribeiro et al. (2016); Shrikumar et al. (2017), which can be influenced by human biases and don't always scale well (as shown by Wang et al. (2019); Rosenfeld (2021)), a wide array of metrics have been introduced to evaluate XAI methods based on these three criteria quantitatively. These metrics are deployed in several studies (see Table 1) to determine *"What XAI method should I (not) use for my problem?"*.

However, this current work in validating XAI methods has two major shortcomings:

**Shortcoming 1: Gaps and inconsistencies in XAI evaluation.** Many studies restrict their analyses to a limited set of design parameters such as modalities, (toy-)datasets, model architectures, methods, and metrics, which all directly impact the performance of XAI methods (we define the first three parameters as *underlying* design parameters, as they directly influence the XAI method). Table 1 demonstrates this fragmented landscape specifically for the domain of computer vision, including discrepancies found across

| XAI Method: Studies: | Attribution | | | | | | | | | | | | | | Attention | | |
|---|---|---|---|---|---|---|---|---|---|---|---|---|---|---|---|---|---|
| | OC | LIME | KS | VG | IxG | GB | GC | SC | C+ | IG | EG | DL | DLS | LRP | RA | RoA | LA |
| Adebayo et al. (2018) ($n_A = 5\ (8)$, $n_E = 2$) | | | | F | F | F | F | | | F | | | | | | | |
| Nie et al. (2019) ($n_A = 2\ (3)$, $n_E = 0$) | | | | F | | F | | | | | | | | | | | |
| Kindermans et al. (2019) ($n_A = 4\ (8)$, $n_E = 1$) | | | | | F | | | | | F | | | | F | | | |
| Ghorbani et al. (2019) ($n_A = 3$, $n_E = 3$) | | | | R | | | | | | R | | R | | | | | |
| Hooker et al. (2019) ($n_A = 3\ (12)$, $n_E = 1$) | | | | F | | F | | | | F | | | | | | | |
| Yang & Kim (2019) ($n_A = 6\ (9)$, $n_E = 3$) | | | | F | F | F | F | | | F | | | | | | | |
| Yeh et al. (2019) ($n_A = 4\ (6)$, $n_E = 2$) | | | R | R | | R | | | | R | | | | | | | |
| Nguyen & Martinez (2020) ($n_A = 3$, $n_E = 4$) | | | | C | C | | | | | C | | | | | | | |
| Chefer et al. (2020) ($n_A = 5$, $n_E = 1$) | | | | | | | F | | | | | | | F | F | F | F |
| Bhatt et al. (2020) ($n_A = 6$, $n_E = 3$) | | | F/R | F/R | F/R | | | | | F/R | | F/R | | | | | |
| Arun et al. (2021) ($n_A = 5\ (8)$, $n_E = 4$) | | | | F | | F | F | | | F | | | | | | | |
| Singh et al. (2021) ($n_A = 9\ (12)$, $n_E = 0$) | C | | | C | C | C | | | | C | | C | C | C | | | |
| Kakogeorgiou et al. (2021) ($n_A = 8\ (12)$, $n_E = 3$) | F/R/C | F/R/C | | F/R/C | F/R/C | F/R/C | F/R/C | | | F/R/C | | F/R/C | | | | | |
| Dombrowski et al. (2022) ($n_A = 5$, $n_E = 1$) | | | | R | R | R | | | | R | | | | R | | | |
| Arras et al. (2022) ($n_A = 6\ (12)$, $n_E = 2$) | | | | F | F | F | F | | | F | | | | F | | | |
| Hesse et al. (2023) ($n_A = 11$, $n_E = 3$) | | F | | | F | | F | | | F | | | | | | F | F |
| Li et al. (2023) ($n_A = 3\ (6)$, $n_E = 3\ (6)$) | | | | F | | | F | | | F | | | | | | | |
| **Ours** * ($n_A = 17$, $n_E = 20$) | F/R/C | F/R/C | F/R/C | F/R/C | F/R/C | F/R/C | F/R/C | F/R/C | F/R/C | F/R/C | F/R/C | F/R/C | F/R/C | F/R/C | F/R/C | F/R/C | F/R/C |

Evaluated Criteria: **F** Faithfulness, **R** Robustness, **C** Complexity
\* Results for image modality

Relative assessment to other XAI methods in study:
🟩 Positive  🟨 Neutral  🟥 Negative

Table 1: Showing gaps and inconsistencies between 17 related studies evaluating XAI methods. Colors coincide with their ranking inside the study depending on the evaluation criteria. $n_A$: Amount of distinct XAI methods. $n_E$: Number of evaluation metrics. If $n_E = 0$, the study was conducted qualitatively.

studies, with some methods, such as GradCAM (GC) (Selvaraju et al., 2017), receiving contradictory assessments depending on the evaluation setup. As a consequence, our current understanding of XAI performance is limited, making it challenging for practitioners to determine a reliable XAI method for their specific use case.

**Shortcoming 2: Individual XAI metrics lack trustworthiness.** Recently, numerous metrics have been proposed to approximate evaluation criteria. However, these metrics are always only a proxy for the criterion they approximate, reflecting the diversity of perspectives on the criterion, and are distinguishable solely in their mathematical formulation. Generally, studies apply one or two metrics to address a criterion (see $n_E$ in Table 2 for the total amount of metrics used for evaluation in each study). Arguably, this is not a reliable measure of success, as these limited subsets can lead to selection bias and overfitting to one metric or one perspective on the criteria. The selection bias between metrics becomes even more severe when mean aggregating across several metrics, which is a common procedure in XAI evaluation (Li et al., 2023; Hesse et al., 2023), as the mean is not robust against outliers, especially when considering only a small set of metrics. Additionally, mean aggregation can conceal variations between metrics, potentially ignoring valuable information. Consequently, rankings and scores of current work lack trustworthiness.

In response to these shortcomings, we developed LATEC: the first comprehensive benchmark tailored for large-scale attribution & attention evaluation in computer vision. LATEC encompasses 17 of the most widely-used saliency XAI methods, including attention-based methods, and evaluates them using 20 distinct metrics (see Figure 1). Notably, LATEC integrates a variety of model architectures, and, to extend the evaluation spectrum beyond traditional 2D images, we included 3D point cloud and volume data, adapting XAI methods and metrics as necessary to suit these modalities. In total, LATEC assesses 7,560 unique combinations.

As LATEC systematically incorporates all vital underlying design parameters affecting XAI methods, we can quantify their effect on XAI methods and perform a more trustworthy and generalizing benchmark, answering

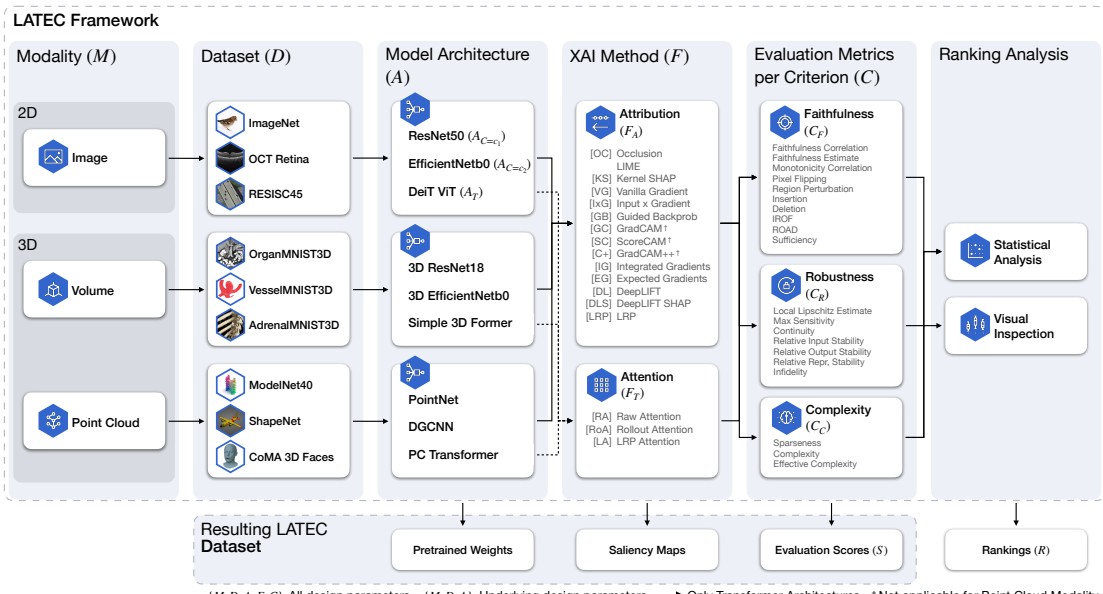

Figure 1: Structure and design parameters of the LATEC framework with the output data for the LATEC dataset of each stage.

Shortcoming 1. However, we can also use LATEC for metric analysis (also called meta-evaluation) to, a priori of the evaluation of the methods, quantitatively validate ranking behavior of metrics, and determine a robust evaluation mechanism, answering Shortcoming 2. Moreover, in support of future research, we've made all intermediate data, including 326,790 saliency maps and 378,000 evaluation scores, as well as the benchmark publicly accessible.

## 2   The LATEC benchmark

The LATEC benchmark includes a framework and a dataset with the rankings as the final output. The framework allows for diverse large-scale studies, structuring the experiments in six stages (see Figure 1), and the dataset provides reference data for evaluation and exploration. As the benchmark is easily extendable and leverages the high-quality dataset for standardized evaluation, it also serves as a foundation for future benchmarking of new XAI methods and metrics. We introduce a mathematical notation for subsequent calculations and formulas. Here, the modality is represented by the symbol $M$.

**Utilized datasets $D$**    For the image modality, we use ImageNet (IMN) (Deng et al., 2009), UCSD OCT retina (OCT) (Kermany et al., 2018) and RESISC45 (R45) (Cheng et al., 2017), the volume modality the Adrenal-(AMN), Organ-(OMN) and VesselMedMNIST3D (VMN) datasets (Yang et al., 2023), and the point cloud modality the CoMA (CMA) (Ranjan et al., 2018), ModelNet40 (M40) (Wu et al., 2014) and ShapeNet (SHN) (Chang et al., 2015) datasets.

**Model architectures $A$**    On each utilized dataset except IMN, where we take pretrained models, we train three models to achieve the architecture-dependent SOTA performance on the designated test set (if available, see Appendix A for a detailed description of the model training and hyperparameters). For the image modality, we use the ResNet50, EfficientNetb0, and DeiT ViT (Touvron et al., 2022) architectures, for the volume modality the 3D ResNet18, 3D EfficientNetb0, and Simple3DFormer (Wang et al., 2022) architectures, and for the point cloud modality the PointNet, DGCNN and PC Transformer (Guo et al., 2021) architectures. The first two architectures are always CNNs $(A_{C=\{c_1,c_2\}})$ and the third is a Transformer $(A_T)$.

**XAI methods $F$**    In total, we include 17 XAI methods, 14 attribution methods $(F_A)$: Occlusion (OC) (Zeiler & Fergus, 2013), LIME (on feature masks) (Ribeiro et al., 2016), Kernel SHAP (KS, on feature masks)

(Lundberg & Lee, 2017), Vanilla Gradient (VG) (Simonyan et al., 2013), Input x Gradient (IxG) (Shrikumar et al., 2017), Guided Backprob (GB) (Springenberg et al., 2015), GC, ScoreCAM (SC) (Wang et al., 2020), GradCAM++ (C+) (Chattopadhay et al., 2018), Integrated Gradients (IG) (Sundararajan et al., 2017), Expected Gradients (EG, also called Gradient SHAP) (Erion et al., 2020), DeepLIFT (DL) (Shrikumar et al., 2017), DeepLIFT SHAP (DLS) (Lundberg & Lee, 2017), LRP (with $\epsilon$-,$\gamma$- and $0^+$-rules depending on the model architecture) (Binder et al., 2016), and three attention methods ($F_T$): Raw Attention (RA) (Dosovitskiy et al., 2021), Rollout Attention (RoA) (Abnar & Zuidema, 2020) and LRP Attention (LA) (Chefer et al., 2021). While the attribution methods are applied to all model architectures, the attention methods can only be applied to the Transformer-based architectures. For comparison reasons, we only consider the original methods without adaptations, as several other works (Hooker et al., 2019; Yang & Kim, 2019) already showed that advancing methods by VarGrad (Adebayo et al., 2018) or SmoothGrad(-squared) (Smilkov et al., 2017) can, in general, improve results. We qualitatively tuned the XAI hyperparameters per dataset (see Appendix B), as also commonly done in practice, to not bias the quantitative evaluation results (see Appendix C for all hyperparameters). However, we observe that most hyperparameters transfer well over all datasets within a modality. We preprocess the saliency maps to ensure fair comparability across all design parameters before computing evaluation scores (see Appendix F for the preprocessing).

**Evaluation metrics and criteria $C$** The evaluation metrics are grouped into three commonly used criteria. The criteria of faithfulness ($C_F$) evaluates if an explanation is following the model's behavior. To approximate faithfulness most metrics validate that the attributed value to an input feature influences the model output in the same amount, implemented e.g. through perturbations based on the attribution values and correlation measurements. The criterion of robustness ($C_R$) evaluates how stable an explanation is, as small changes in the input space should not lead to large changes in the saliency map. To validate this criterion, metrics measure different types of relative changes in the saliency map induced by perturbations in the input space or the interchange with similar input samples. The criterion of complexity ($C_C$) evaluates if an explanation is concise and humanely understandable. To quantify complexity, metrics are leveraging measurements such as entropy or the Gini-coefficient. In total, we utilize 20 evaluation metrics, of which 10 evaluate faithfulness, seven robustness, and three complexity (see Appendix D for a description of every metric). All metrics are assessed for their adequacy; only well-established metrics specifically proposed for image data (i.e. ignoring metrics proposed for specific synthetic data or other modalities like natural language) and their respective criteria are selected for evaluation. We tune the hyperparameters per dataset as some depend on dataset properties (see Appendix subsection D.2 for all parameters).

**3D adaption** While several XAI methods and metrics in LATEC are independent of the input space dimensions, we extended many of them to 3D volume and point cloud data, building upon the implementations for image data by Kokhlikyan et al. (2020) and Hedström et al. (2023). We describe the adaption process for all respective XAI methods and metrics in Appendix E and show illustrative saliency maps. All adaptations are tested for their coherency.

**Ranking computation** As the nominal evaluation scores ($S$) have no semantic meaning and their scales differ between utilized datasets, we analyze the XAI methods based on their ranking ($R$). To this end, we compute the median evaluation score over all observations on the dataset level and rank the methods accordingly, achieving more robust results. When calculating ranking statistics, no distributional assumptions are required, and they offer robustness against outliers. These characteristics are crucial given that metric distributions are often highly skewed. However, this approach does not capture the extent of variation among score distributions. To address this, we present the distribution of all scores prior to ranking in Appendix O. See Appendix G for a detailed flow chart of how we get from evaluation scores to rankings. We utilize these rankings in the subsequent metric analysis and benchmark and refer to Appendix C for all implementation details and Appendix B for more information about the dataset and ranking computation.

## 3 Metrics analysis

**How severe is the risk of metric selection bias in XAI evaluation?** In Shortcoming 2, we describe a risk of selection bias due to evaluating one or a few metrics. In our metrics analysis, we first aim to provide empirical evidence for this risk. A first exploratory analysis quickly supports the hypothesis, as we encounter

**a.** Ranking-(Dis)agreement between Metrics ($R_{c, F=\{LIME,GC,IG,DLS\}, d=ImageNet, a=ResNet50}$)

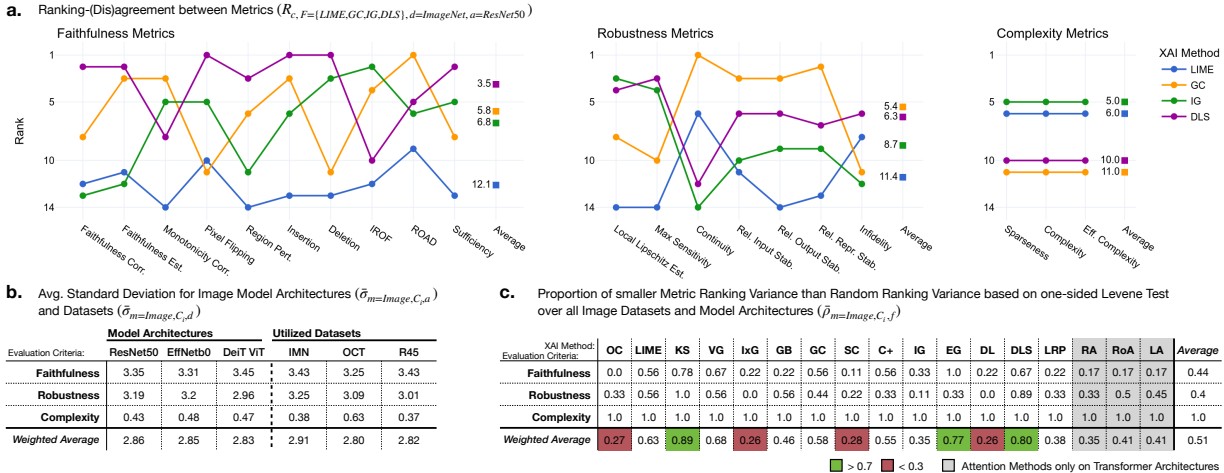

**b.** Avg. Standard Deviation for Image Model Architectures ($\bar{\sigma}_{m=Image,C_i,a}$) and Datasets ($\bar{\sigma}_{m=Image,C_i,d}$)

| Evaluation Criteria: | Model Architectures | | | Utilized Datasets | | |
|---|---|---|---|---|---|---|
| | ResNet50 | EffNetb0 | DeiT ViT | IMN | OCT | R45 |
| **Faithfulness** | 3.35 | 3.31 | 3.45 | 3.43 | 3.25 | 3.43 |
| **Robustness** | 3.19 | 3.2 | 2.96 | 3.25 | 3.09 | 3.01 |
| **Complexity** | 0.43 | 0.48 | 0.47 | 0.38 | 0.63 | 0.37 |
| *Weighted Average* | 2.86 | 2.85 | 2.83 | 2.91 | 2.80 | 2.82 |

**c.** Proportion of smaller Metric Ranking Variance than Random Ranking Variance based on one-sided Levene Test over all Image Datasets and Model Architectures ($\bar{\rho}_{m=Image,C_i,f}$)

| XAI Method: Evaluation Criteria: | OC | LIME | KS | VG | IxG | GB | GC | SC | C+ | IG | EG | DL | DLS | LRP | RA | RoA | LA | *Average* |
|---|---|---|---|---|---|---|---|---|---|---|---|---|---|---|---|---|---|---|
| **Faithfulness** | 0.0 | 0.56 | 0.78 | 0.67 | 0.22 | 0.22 | 0.56 | 0.11 | 0.56 | 0.33 | 1.0 | 0.22 | 0.67 | 0.22 | 0.17 | 0.17 | 0.17 | 0.44 |
| **Robustness** | 0.33 | 0.56 | 1.0 | 0.56 | 0.0 | 0.56 | 0.44 | 0.22 | 0.33 | 0.11 | 0.33 | 0.0 | 0.89 | 0.33 | 0.33 | 0.5 | 0.45 | 0.4 |
| **Complexity** | 1.0 | 1.0 | 1.0 | 1.0 | 1.0 | 1.0 | 1.0 | 1.0 | 1.0 | 1.0 | 1.0 | 1.0 | 1.0 | 1.0 | 1.0 | 1.0 | 1.0 | 1.0 |
| *Weighted Average* | 0.27 | 0.63 | 0.89 | 0.68 | 0.26 | 0.46 | 0.58 | 0.28 | 0.55 | 0.35 | 0.77 | 0.26 | 0.80 | 0.38 | 0.35 | 0.41 | 0.41 | 0.51 |

■ > 0.7    ■ < 0.3    □ Attention Methods only on Transformer Architectures

Figure 2: **a.** Ranking of four XAI methods based on all evaluation metrics of each criterion for one specific set of design parameters. **b.** Average standard deviation per model architectures (based on Equation 1) and utilized datasets (Equation 2) for the image modality. Weighted average per column is based on the number of metrics per criterion (i.e. $n_{C_F} = 10$, $n_{C_R} = 7$, and $n_{C_C} = 3$). **c.** Proportion of accepted one-sided Levene-Tests for significantly smaller ranking variance compared to the variance of an entire random ranking (based on Equation 3). Higher values show higher agreement between metrics. The weighted average is based on the number of metrics per criterion.

strong ranking disagreement between metrics for various combinations of underlying design parameters, which increases the risk of metric selection bias. We define ranking agreement as the consensus among metrics belonging to one criterion about the rank of one XAI method when evaluating and subsequently ranking this method against all other XAI methods. Consequently, disagreement in ranking is defined through high variance between the determined ranks of the metrics for one XAI method. For example, Figure 2 (a.) demonstrates the ranking behavior of four selected XAI methods for one selection of underlying design parameters ($R_{c,F_a=\{\text{LIME,GC,IG,DLS}\},d=\text{ImageNet},a=\text{ResNet50}}$). The line charts show how each metric ranks the four XAI methods in comparison to all other XAI methods for this specific selection of underlying design parameters, with the mean aggregated average rank over the metrics of one criterion to the right. For faithfulness, we observe high disagreement between metrics in their ranking of GC and IG, mainly agreeing metrics in the ranking of DLS (with IROF being a noticeable outlier) and agreeing metrics in the ranking of LIME. The inquiry emerges as to whether the risk of selection bias is generally present in certain combinations of underlying design parameters or is uniformly distributed across them. To this end, we computed the standard deviation (SD) between metric rankings for each modality, criteria and model architecture ($\bar{\sigma}_{m,C_i,a}$), mean aggregated over XAI methods and datasets, as well as for each modality, criteria and dataset ($\bar{\sigma}_{m,C_i,d}$), mean aggregated over XAI methods and model architecture:

**Average SD between metrics** (Figure 2 (b.))
Using the notation introduced in section 2 and SD as $\sigma(\cdot)$. Per model architecture ($a$):

$$\bar{\sigma}_{m,C_i,a} = \frac{1}{|D||F|} \sum_{d\in D, f\in F} \sigma(R_{C_i,a,f,m,d}) \quad \forall m \in M, \ \forall a \in A, \ \forall C_i \in \{C_F, C_R, C_C\}, \tag{1}$$

and dataset ($d$):

$$\bar{\sigma}_{m,C_i,d} = \frac{1}{|A||F|} \sum_{a\in A, f\in F} \sigma(R_{C_i,a,f,m,d}) \quad \forall m \in M, \ \forall d \in D, \ \forall C_i \in \{C_F, C_R, C_C\} \tag{2}$$

Figure 2 (b.) shows for the imaging modality ($\bar{\sigma}_{m=\text{Image},C_i,a}$ and $\bar{\sigma}_{m=\text{Image},C_i,d}$, see Appendix H for the other two modalities) that the average SD is generally stable between model architectures or datasets within each

evaluation criterion. Thus, we can conclude that there is no single model architecture, modality, or dataset choice that has a substantial effect on the disagreement between metric rankings of all XAI methods.

Now that we can rule out the general influence of underlying design parameters, we quantify how strong the risk of metric disagreement is in general and if there is a difference between XAI methods. To this end, we utilize a one-sided Levene's Test (Levene, 1960), testing if the rank-variance of a set of metrics is significant lower than the variance of a random rank distribution, which can be analytically inferred. We compute this test for all sets of metrics on every possible combination of design parameters and mean aggregate over model architecture and datasets. Figure 2 (c.) shows for the imaging modality the resulting proportion of accepted tests ($\bar{\rho}_{m,C_i,f}$) of each criteria and XAI method, i.e. tests where the computed p-value — the probability that an observed effect occurs by chance — is below the significance level $\alpha$:

> **Proportion of accepted Levene tests** (Figure 2 (c.))
> With significance level $\alpha = 0.1$, p-value $PV_{\text{Le}}(\cdot)$, indicator function $\mathbb{1}[\cdot]$ and variance $\sigma^2(\cdot)$.
>
> $$\bar{\rho}_{m,C_i,f} = \frac{1}{|D||A|} \sum_{d \in D, a \in A} \mathbb{1}[PV_{\text{Le}}(\sigma^2(R_{C_i,f,m,a,d})) < \alpha] \;\; \forall m \in M, \forall f \in F, \forall C_i \in \{C_F, C_R, C_C\} \quad (3)$$

By computing the weighted average proportion across criteria at the bottom, we indeed observe strong variations between the XAI methods. Specifically for KS, EG, and DLS, in a large majority of cases, metrics agree, while for OC, IxG, SC, and DL only in about $\sim 27\%$ of the cases variance in metric ranking is significantly lower than random ranking. Concluding, our findings reveal that metrics disagree and agree in varying degrees depending on the XAI method.

**Why do metrics disagree?** We have established that all metrics approximate similar criteria, differing primarily in their interpretation and mathematical formulation. Although these differing perspectives mainly agree with their rankings, our study reveals that variations in mathematical formulation can significantly contribute to metric disagreement. Our prior research indicates that the extent of disagreement between metrics is significantly influenced by the chosen XAI method. This dependency emerges as a critical factor in why certain metrics may favor or disadvantage specific XAI methods due to their mathematical structures. Moreover, our further experiments demonstrate that metrics particularly diverge in their rankings of XAI methods in terms of faithfulness, especially when the mechanism used for evaluation and the mechanism for computing the explanation (i.e. saliency map) are closely related.

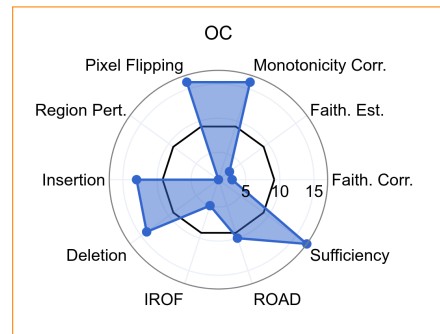

Figure 3: Differences between faithfulness metrics in ranking OC based on $R_{C_f, f=\text{OC}, d=\text{ImageNet}, a=\text{ResNet50}}$. The black circle indicates the overall average rank.

In Figure 2 (c.) we can observe for OC that the SD between faithfulness ranks is never significantly smaller than the deviation of a random ranking (see exemplary Figure 3 for all individual faithfulness metric ranks of OC for one set of design parameters). A primary cause of this notable metric disagreement is OC's alignment with the Region Perturbation metric. Both OC and Region Perturbation involve perturbing a larger region of the input image with a baseline value, utilizing either a fixed kernel in OC or a set of pixels determined by ordered attribution values in Region Perturbation. When the set size for Region Perturbation matches the OC kernel size, and there is no overlap of the sliding kernels, the metric operationally mirrors the XAI method. However, our study indicates that even when the set size does not align, the metric still tends to favor the method. Figure 3 highlights OC's high rankings when evaluated using Region Perturbation and lower rankings with metrics that employ finer, incremental pixel-level perturbations, such as Pixel Flipping, Insertion, or Deletion. This evaluation bias stems from OC's inherent limitation of attributing to entire regions rather than individual pixels.

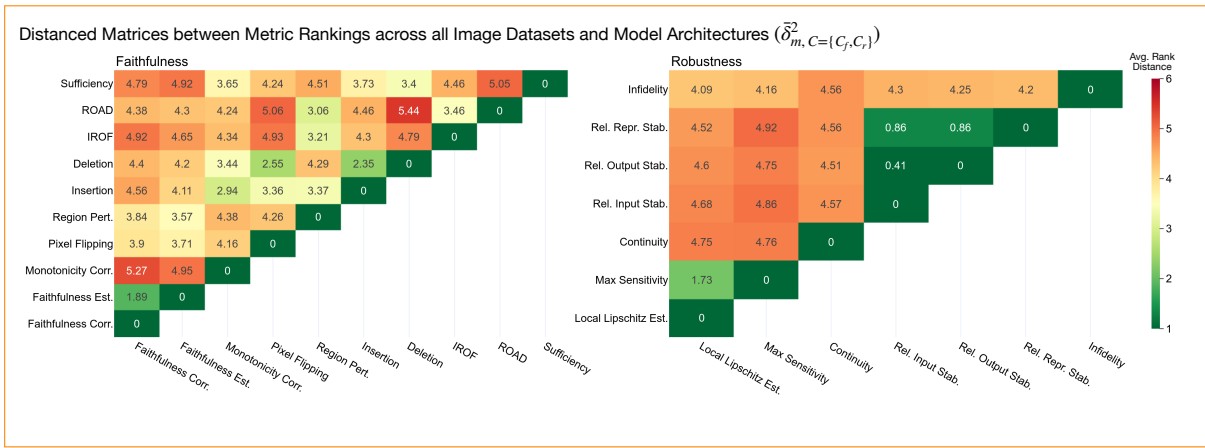

Figure 4: Average Euclidean distance matrix between metric ranks for the faithfulness and robustness criteria across all XAI methods, image datasets, and model architectures (see Equation 4). The green color coincides with similar ranking metrics, and the red with dissimilar. See Appendix Figure 11 for distance matrices for each model architecture individual.

We observe a dependency not only between specific XAI methods and metrics but also among metrics that utilize similar evaluation mechanisms or those designed to specifically address the shortcomings of other metrics. This scenario poses a risk for selection bias, as we observe that several related studies, such as Li et al. (2023), predominantly select such metrics with similar methodologies and consequently similar ranking behavior.

Figure 4 illustrates the average Euclidean ranking distance ($\bar{\delta}_{m,c}$) between all metrics across image datasets and model architectures (see Appendix I for each model architectures individual):

> **Average Euclidean distance between metrics** (Figure 4)
>
> $$\bar{\delta}_{m,c} = \frac{1}{|D||A||F|} \sum_{d \in D, a \in A, f \in F} \sqrt{(R_{m,c,d,a,f} - R_{m,c,d,a,f})^2} \quad \forall m \in M, \forall c \in C \qquad (4)$$

In assessing faithfulness, metrics that involve incremental pixel perturbation (Pixel Flipping, Insertion, Deletion) and those that correlate attribution values with predicted logits (Faithfulness Correlation, Faithfulness Estimate) tend to rank more similarly. Conversely, metrics designed to mitigate specific limitations of these methods display notably different rankings. For instance, incrementally inserting or deleting pixel-based methods may generate out-of-distribution examples for the model, leading to highly uncertain or even random predictions. This issue is addressed by the ROAD metric, which consequently ranks distinctly from Deletion and Pixel Flipping. Similarly, the Monotonicity Correlation metric, which evaluates the correlation between the absolute attribution values and the uncertainty in probability estimates, addresses shortcomings in Faithfulness Correlation and Estimate, leading to divergent rankings. Regarding robustness metrics, variations are more consistent. There is a notable similarity in rankings between the Local Lipschitz Estimate and Max Sensitivity, both of which measure relative changes when inputs are slightly altered, as well as between the Relative Stability metrics. Disparities in rankings among metrics may also arise from variations in the tuning of hyperparameters.

In summary, our study demonstrates that the variation in metric rankings for a given XAI method can be attributed to the similarity or dissimilarity in the mathematical mechanisms employed by the metrics themselves, as well as between the metrics and the underlying XAI method. However, in scenarios where there is substantial disagreement among metrics, selection biases may emerge if only a limited subset of metrics—those that potentially employ similar mechanisms—is considered. This underscores the importance

of incorporating a diverse array of metrics to ensure an accurate approximation of a criterion, independent of the mathematical mechanisms involved.

**How can we reliably benchmark and find robust trends within agreeing and disagreeing metrics?** The current practice in XAI evaluation is to employ a small set of metrics and subsequently mean aggregate over the normalized scores to increase the generalization of results and simplify data analysis in big datasets (see e.g. Li et al. (2023); Hedström et al. (2023); Hesse et al. (2023)).

Nevertheless, this current procedure exhibits three major limitations. First, current practice aggregates only over small sets of metrics, which comes with an increased risk of biases due to a high dependence on metric selection and individual metric behavior, as especially the mean is not robust to outliers in such small samples Our metric analysis demonstrates this risk of selection bias extensively. Such bias can compromise the integrity of the evaluation, potentially substantially lowering its validity. We start addressing the first limitation by including the to-date largest scale of diverse and relevant metrics, that exhibit diversity in their implementation and interpretation of the respective criteria, minimizing the risk of selection bias and the influence of outliers. The metrics are systematically chosen for their relevance to the modality and criterion, such as excluding metrics specific to natural language data.

Second, the aggregation of normalized scores can be flawed due to unbounded metrics, inconsistent interpretations, and sensitivity to outliers and distribution skewness. In such a large-scale study, aggregation is necessary as a mechanism of abstraction to extract meaningful insights from this large extent of data (i.e. 378.000 evaluation scores in our case). In addition, we are also interested in detecting generalizing trends, i.e. rankings that generalize across design parameters such as e.g. all datasets of one modality. Current studies use the aggregation of normalized scores, which is known to have several flaws: metrics can be unbounded, interpretations of measurements may differ across metrics even when scaled similarly, and the aggregation can be influenced by outliers and skewed distributions (also shown by Colombo et al. (2022)). We address this limitation by employing a "rank-then-aggregate" scheme, which is already well established for large-scale evaluation of model performance (Maier-Hein et al., 2018; Colombo et al., 2022; Rosset et al., 2005) as ranking statistics are more robust against skewed distributions and outliers compared to the raw scores. It can also be argued that rankings have higher interpretable value because they simplify comparison by clearly indicating the relative performance or position of methods, eliminating the need to understand varied scales or units (also argued by Rosset et al. (2005)). Although it is not without its imperfections, mean aggregation possesses the ability to highlight only strong trends across metrics. For a method to achieve consistently high or low average rankings, the metrics assessing a particular criterion must largely agree in their evaluations (such as faithfulness in the cases of LIME and DLS shown in Figure 2(a)). Given the fact that rankings have inherent upper and lower bounds, substantial disagreement among them, indicated by a high SD, will naturally converge the average rank to a more intermediate (i.e. mean) average rank (as the "Average" faithfulness for GC and IG in Figure 2 (a.)). We leverage this dynamic by focusing only on strong trends where there is consensus across metrics.

At last, current studies discard the standard deviation during aggregation, which contains crucial information about a method's performance. XAI methods that receive intermediate average scores or ranks may attain such a categorization either because all metrics consistently agree or because they exhibit significant disagreement. The SD is necessary to resolve such ambiguities as it communicates the uncertainty behind an aggregation. We believe that understanding why metrics disagree and the situations in which this occurs is vital for evaluating XAI. To determine if a variance is large in general, we deploy the Levene test in Equation 3 to test if a variance between metric ranks is significantly smaller than the variance of a random ranking. In the XAI benchmark however, we are only interested in comparing the SD between XAI methods. To this end, we use the quantiles of the SD distributions to determine threshold values within each evaluation criterion.

To ensure that the aggregated mean is not disproportionately influenced by metrics that employ similar evaluation mechanisms, we employ distance matrices from Figure 4 (or Appendix I for more refined weighting). This approach allows us to compute a weighted aggregated mean, down-weighting the influence of related metrics. In our experiments, however, we observed that the impact of this adjustment is minimal, as related metrics do not obligatory have to rank similarly. Thus we compute in Table 2 the mean aggregated average

rank ($\bar{R}_{\bar{C},m,f}$) for each XAI method, modality, and criterion. When mean aggregating ranks, the subscripts of $\bar{R}$ denote the design parameters the ranking value is shown while we aggregate across the remaining. Alongside, we show the average SD ($\bar{\sigma}_{C_i,m,f}$) between metrics, to assess the reliability of each average rank:

**Average SD between metrics for each $\bar{R}_{\bar{C},m,f}$** (Table 2 (b.))

$$\bar{\sigma}_{C_i,m,f} = \frac{1}{|D||A|} \sum_{d \in D, a \in A} \sigma(R_{C_i,a,f,m,d}) \quad \forall f \in F, \ \forall m \in M, \ \forall C_i \in \{C_F, C_R, C_C\}, \tag{5}$$

Our initial observations validate the mathematically grounded hypothesis that methods ranked at the extremes exhibit smaller SDs. For example, the SD between faithfulness metrics for OC is 5.31 with an average rank of 11.5, and for KS it is 3.07 with a average rank of 16.5. This supports the robustness of our subsequent XAI benchmark analysis.

**Encountered pitfalls in XAI metric application.** During our analysis, we encountered several critical pitfalls of current metric application in XAI, to our knowledge not discussed before. Suspiciously, Figure 2 (c.) indicates almost no disagreement between complexity rankings ($\bar{\rho}_{m,C_C,f}$). By investigating complexity metrics further, we observed that gradient and Deep Taylor Decomposition principle-based methods (IxG, IG, DL, and LRP, see Table 2) are ranked as significantly less complex compared to the CAM and attention methods. In our opinion, this observation is counterintuitive when comparing the complexity rankings to the saliency maps in Appendix B, based on which we would classify CAM and attention methods as more clearly arranged and less noisy. While all three complexity metrics are also explicitly proposed for image data, we notice that they all treat each pixel, voxel, or point independently of each other, ignoring locality and favoring methods that attribute to the smallest set of single pixels. As this approach possibly transfers to low dimensional images such as MNIST (Lecun et al., 1998) or CIFAR-10 (Krizhevsky, 2009), the image datasets the three metrics are originally presented on, we hypothesize that it may not be effective with higher-dimensional inputs as observed in our study. Consequently, it is expected that techniques such as LRP would be highly regarded due to their emphasis on filtering the significance of individual pixels, in contrast to CAM methods (GC, SC, C+) that attribute importance to broader local regions. Whether these XAI methods are less complex and more human-understandable on computer vision modalities is debatable and *subsequent complexity evaluation results should be interpreted with caution.* While all metrics are theoretically very well founded, we encountered several smaller pitfalls during our evaluation, which can be found in Appendix L. However, even if a metric fails for a specific combination of design parameters, we still include them in the benchmark, as the determination of such failure cases is pivotal to future work in metric analysis.

## 4 XAI benchmark results

Upon establishing a robust framework for evaluation, we address Shortcoming 1 by decomposing it into four essential sub-questions. The first two questions aim to bridge gaps in research due to limited design parameters, while the last two focus on resolving inconsistencies by identifying key trends in XAI method performance. Every sub-question includes a set of **four main findings (1-4)**, each of which consists of an *observation* followed by a *recommendation*. For more fine-grained assistance in selecting the most appropriate XAI methods, we refer to detailed ranking tables (see also Appendix K) and the LATEC dataset.

**How is XAI method performance dependent on underlying modalities and datasets?** Although, modalities in computer vision share attributes like locality, spatial structure, and associated feature descriptors, they differ in aspects such as dimensionality and representation forms (e.g., grid vs. points). If these differences indeed influence XAI performance, the resulting inconsistencies are largely overlooked in the current literature. To quantify such an effect on XAI methods, we average ranks of XAI methods within modalities and evaluation criteria $\bar{R}_{\bar{C},m,f}$ in Table 2 (a.) alongside the SD between metrics in (b.), revealing new patterns in ranking behavior of various methods. **(1)** Between modalities, only a few methods achieve generalizing results over all modalities, even for one criterion. This observation is most applicable to the methods IG and VG, which also show a standard SD between metrics. We generally recommend that users

**a.** Average Rank of Metrics per Criterion ($\bar{R}_{\bar{C},m,f}$)

| Eval. Criteria: | Faithfulness | | | Robustness | | | Complexity | | |
|---|---|---|---|---|---|---|---|---|---|
| Modality: | Image | Volume | Point Cloud | Image | Volume | Point Cloud | Image | Volume | Point Cloud |
| OC | 11.5 | 9 | 12 | 12.5 | 9.5 | 12 | 9 | 7.5 | 6.5 |
| LIME | 16.5 | 13 | 2 | 16 | 15 | 14 | 7 | 9.5 | 9 |
| KS | 16.5 | 12 | 3 | 17 | 17 | 12 | 12.5 | 12 | 10 |
| VG | 13 | 14 | 8.5 | 4 | 4 | 6.5 | 11 | 14 | 8 |
| IxG | 9.5 | 5 | 5 | 14.5 | 11.5 | 9.5 | 4 | 2 | 3 |
| GB | 7.5 | 10 | 1 | 6.5 | 8 | 12 | 5 | 6 | 5 |
| GC | 6 | 17 | - | 2.5 | 13 | - | 14 | 9.5 | - |
| SC | 4.5 | 11 | - | 10 | 15 | - | 10 | 16.5 | - |
| C+ | 11.5 | 15.5 | - | 5 | 11.5 | - | 15.5 | 16.5 | - |
| IG | 4.5 | 2 | 4 | 10 | 4 | 9.5 | 3 | 3.5 | 4 |
| EG | 1 | 3 | 7 | 2.5 | 1 | 6.5 | 17 | 7.5 | 11 |
| DL | 7.5 | 5 | 6 | 12.5 | 7 | 8 | 2 | 3.5 | 2 |
| DLS | 2 | 5 | 12 | 6.5 | 9.5 | 5 | 6 | 5 | 6.5 |
| LRP | 15 | 15.5 | 10 | 14.5 | 15 | 2 | 1 | 1 | 1 |
| RA | 14 | 8 | 8.5 | 1 | 6 | 3.5 | 8 | 11 | 12 |
| RoA | 9.5 | 7 | 12 | 8 | 2 | 3.5 | 15.5 | 13 | 14 |
| LA | 3 | 1 | 14 | 10 | 4 | 1 | 12.5 | 15 | 13 |

Per modality: ▮ Top 1  ▯ Top 2-4  ▯ Bottom 2-4  ▮ Bottom 1

**b.** Average Standard Deviation between Metrics per Criterion ($\bar{\sigma}_{\bar{C},m,f}$)

| Eval. Criteria: | Faithfulness | | | Robustness | | | Complexity | | |
|---|---|---|---|---|---|---|---|---|---|
| Modality: | Image | Volume | Point Cloud | Image | Volume | Point Cloud | Image | Volume | Point Cloud |
| OC | 5.31 | 3.47 | 4.04 | 4.16 | 3.75 | 3.41 | 0.31 | 1.57 | 0.78 |
| LIME | 3.94 | 3.36 | 3.34 | 3.71 | 3.80 | 2.71 | 1.20 | 1.03 | 1.25 |
| KS | 3.07 | 3.32 | 3.16 | 3.49 | 3.72 | 1.90 | 0.63 | 1.35 | 0.47 |
| VG | 3.40 | 4.32 | 3.29 | 2.56 | 5.06 | 2.55 | 0.73 | 0.85 | 0.85 |
| IxG | 4.95 | 4.32 | 3.12 | 4.71 | 4.62 | 2.91 | 0.31 | 1.91 | 0.63 |
| GB | 5.10 | 4.67 | 3.62 | 3.51 | 4.61 | 2.04 | 0.16 | 1.14 | 0.57 |
| GC | 3.39 | 2.77 | | 3.98 | 3.32 | | 0.47 | 1.02 | |
| SC | 4.35 | 4.31 | | 4.30 | 4.00 | | 0.94 | 2.30 | |
| C+ | 3.06 | 4.10 | | 4.64 | 4.20 | | 0.70 | 1.77 | |
| IG | 4.58 | 4.21 | 3.87 | 4.46 | 3.44 | 4.84 | 0.47 | 1.00 | 0.94 |
| EG | 3.25 | 3.64 | 3.71 | 4.06 | 4.46 | 3.05 | 0.63 | 0.73 | 0.42 |
| DL | 4.35 | 3.11 | 3.33 | 5.12 | 4.84 | 3.03 | 0.31 | 0.73 | 0.16 |
| DLS | 3.36 | 2.61 | 2.45 | 3.23 | 4.04 | 1.42 | 1.04 | 0.79 | 0.83 |
| LRP | 3.41 | 4.06 | 2.69 | 4.01 | 6.44 | 0.80 | 0.00 | 0.78 | 0.47 |
| RA | 4.57 | 4.86 | 5.10 | 4.39 | 4.47 | 3.89 | 0.90 | 2.90 | 0.00 |
| RoA | 4.60 | 5.04 | 4.46 | 4.49 | 3.25 | 3.95 | 0.43 | 2.09 | 0.47 |
| LA | 5.16 | 4.33 | 5.27 | 4.05 | 3.41 | 3.79 | 0.47 | 1.75 | 0.47 |

Per Evaluation Criteria: ▮ < 0.15 Quantile  ▮ > 0.85 Quantile

Table 2: **a.** The table shows all values of $\bar{R}_{\bar{C},m,f}$, the average rank for each combination of evaluation criterion, modality and XAI method. For example, the top left entry $\bar{R}_{\bar{C}_F,m=\text{Image},f=\text{OC}}$ is average over $(n_{C_F} = 10) * (n_D = 3) * (n_A = 3) = 90$ ranks. Coloring coincides with top and bottom positions as point cloud rankings are of length 14 and all others are of length 17. See Appendix Table 10 for table with all datasets, Table 11 with only CNN model architectures and Table 12 only Transformer architectures. **b.** Shows the respective average SD between metrics for the ranks reported in (a.) (see Equation 5). Green coloring coincides with the upper 0.15 and red with the lower 0.85 quantiles of each evaluation criterion.

should make their XAI method selection depending on the modality. **(2)** The extended table in Appendix K shows that ranking disparities between datasets within individual modalities are minimal. This suggests that a method selected for one dataset can transfer well to others if dimensionality and characteristics are not too distinct. **(3)** For some methods, we observe a high similarity between performance on the image and volume modality. In particular, both linear surrogate methods (LIME, KS) underperform on image and volume compared to the lower dimensional point cloud modality in terms of faithfulness. On these modalities, their performance also strongly depends on the suitability of the feature mask computed via a grid or super-pixels, which is very time-consuming to fine-tune for single observations. Concluding, we advise against using them for high-dimensional and complex relationships. **(4)** CAM methods (GC, SC, C+) always perform better on image than on volume data. When comparing the saliency maps between both modalities, we observe that the volume-based maps are much coarser (i.e. more "blocky") and less focused. We attribute this observation to less accurate latent model representations and subsequent up-sampling in 3D compared to 2D space, subsequently not recommending them for volume data. The SD of CAM methods is on average lower between faithfulness than between robustness metrics. *In summary, performance relies generally on the modality but not on the dataset.*

**How does XAI method performance depend on the underlying classifier?** We investigate this frequently neglected design parameter by focusing on the aspects of model performance and architecture type. Model performance evaluation might be underemphasized in current research if, even after adequate training, XAI performance remains dependent on model performance. To this end, we compute in Figure 5 (a.) the average Pearson correlation coefficient ($\overline{PCC}$) between the aggregated ranks per criteria and the F1 test scores ($F1$) of the models per modality and criterion:

---

**Correlation between ranks and model performance** (Figure 5 (a.))
With Pearson correlation coefficient $PCC(\cdot)$ and F1 score.

$$\overline{PCC}_{\bar{C},m} = \frac{1}{|D||F|} \sum_{d \in D, f \in F} PCC(R_{\bar{C},m,d,f,A},\ \text{F1}_{m,d,A}) \quad \forall m \in M \tag{6}$$

---

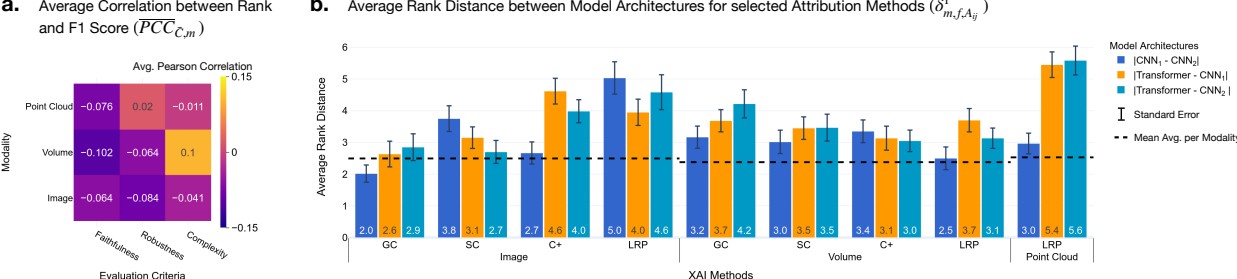

Figure 5: **a.** Correlation between ranks of XAI methods and model F1 test scores, aggregated for each modality and criterion combination (see Equation 6). **b.** Average distance between ranks on different model architectures for CAM methods and LRP, as their distance is well above the average (see Equation 7). See Appendix M for all XAI methods and modalities.

**(1)** Even for well-trained models, we observe a small trend that a better F1 score correlates with better faithfulness and robustness rank on image and volume modalities ($\overline{PCC}$ between -0.06 and -0.1, correlation is negative as a higher F1 scores correlate with a lower, i.e. better, XAI evaluation rank.). The overall observed weak correlation between model performance and evaluation rank raises questions about whether optimizing model performance (in terms of F1 score) to a high degree of precision would yield significant alterations in the saliency map, which ultimately would lead to improvement in the evaluation criteria. Thus, in tasks where several models with high performance but different properties are available for XAI analysis (e.g., for knowledge/scientific discovery), we would additionally recommend taking into account other, more practical, criteria, such as the use of attention methods or operational limitations.

Recent research around Vision-Transformers has shown distinct differences in their learning dynamics (Raghu et al., 2022; Park & Kim, 2022), robustness (Zhou et al., 2022) or latent representations (Wang et al., 2023) compared to classical CNNs. As many of their mechanisms (e.g., global processing, lack of inductive biases, (self-)attention, negative activations) can theoretically also affect attribution methods, we want to test if similar distinctions between Transformer and CNN architectures can also be detected for the performance of attribution methods. To this end, we analyze the distance of ranks between each model architecture for each attribution method (see Figure 5 (b.) and Appendix M for all attribution methods and modalities and Appendix N for differences in ranking order):

> **Average absolute rank distance between model architectures** (Figure 5 (b.))
>
> $$\bar{\delta}^1_{m,f,A_{ij}} = \frac{1}{|D||C|} \sum_{d \in D, c \in C} |R_{m,c,f,a_i} - R_{m,c,f,a_j}| \quad \forall m \in M, \ \forall c \in C, \ \forall f \in F, \ \forall \{a_i, a_j \neq a_i\} \in A \quad (7)$$

**(2)** The average rank distance $(\bar{\delta}^1_{m,f,A_{ij}})$ does not change substantially between CNN and Transformer architectures for most attribution methods (except for CAM methods and LRP), which is mainly centered around the mean value of the respective modality. This indicates that almost all attribution methods do not receive significantly different evaluation scores depending on the underlying architecture and we can not support the hypothesis that attribution methods behave fundamentally differently on Transformer architectures compared to CNNs. **(3)** CAM methods generally show a higher rank distance between architectures, which could be attributed to differences in latent representations of the models, as the semantics captured in the last convolutional or cls-token layers do not have to coincide between models. Thus, we recommend increasing the robustness by averaging the activation map of several hidden layers, which has shown effective in application (Gildenblat, 2024), but can lead to less accurate saliency maps. **(4)** LRP shows additionally high dissimilarity between CNN and Transformer architectures, especially for the 3D modalities. We use the recommended $\gamma$- and $\epsilon$-rules in LRP for the CNN models. However, on Transformer architectures, LRP does not preserve the conservation rule and only works with the $0^+$-rule (see Chefer et al. (2021) for a detailed

explanation). Both implemented changes to LRP bias the relevance computation, which consequentially impacts its performance on Transformer architectures. Thus, we recommend using LA instead of LRP as a relevance-based method on Transformer architectures, as it leverages the Transformer-inherent attention and performs much better regarding faithfulness and robustness. *Summarizing, XAI performance just weakly depends on the model performance and the model architecture only significantly influences CAM and LRP methods.*

**What behavioral similarities exist among XAI methods?** To resolve inconsistencies in current research for method selection, our analysis of XAI behavior focuses on two key aspects: similarities among methods and distinct performance trends. Similarity is important in method selection because choosing a heterogeneous set of XAI methods includes different perspectives on the explanation, which is often advantageous in application. Specifically, we analyze the similarity between single methods and the subgroups of attention and attribution methods, obtaining findings 1-4, answering our main question. Figure 6 (a.) shows the correlation in ranking between XAI methods, indicating their relative similarity:

> **Ranking correlation between XAI methods** (Figure 6 (a.))
> With Pearson correlation coefficient $PCC$.
>
> $$PCC_{F_{ij}} = PCC(R_{\bar{C},M,D,A,f_i}, R_{\bar{C},M,D,A,f_j}) \quad \forall f_i, f_j \in F \tag{8}$$

**(1)** We observe that methods belonging to methodological similar groups are positively correlated: Linear surrogate methods (LIME, KS), CAM methods (GC, SC, C+), and attention methods (RA, RoA, LA). Also, CAM and attention methods are slightly positively correlated, indicating their similar attributing to local regions. We would advise not restricting access to such methodological subgroups to preserve method diversity in application. **(2)** Contrarily to other method subgroups, the Shapely value approximating SHAP methods (EG, KS, and DLS) are not correlated. Also, their performances in Table 2 differ extensively. This observation is consistent with the results of Molnar et al. (2022), which are, however, not in the context of XAI evaluation. Therefore, it is advisable not to select a single SHAP method with the expectation of achieving similar results to others but rather to employ multiple such methods. **(3)** CAM and attention methods negatively correlate with IxG, GB, IG, and DL, which contrarily attribute to single pixels, resulting in more fine-grade saliency maps. Interestingly, we observe a very strong positive correlation between IG/IxG, DL/IxG, and IG/DL, indicating very homogeneous behavior between the methods, even though they are based on different mathematical mechanisms. We would strongly recommend mixing such single-pixel and local-region attributing methods, not only for the diversity in visualization but also because of their different performance in evaluation.

Due to the success of Transformers, attention methods are one of the most emerging subgroups of XAI methods. This raises a pressing question for users: should they exclusively use Transformer-based models for attention methods $(F_T)$, or can architecture-independent attribution methods $(F_A)$ still provide equal or superior explanations? **(4)** When comparing the average ranking $\bar{R}_{\bar{C},F=\{F_A,F_T\}}$ between both groups for all criteria, we observe in Figure 6 (b.) a large difference in complexity and a smaller difference in robustness while the difference in faithfulness is insignificant. The comparatively high robustness of attention methods extends across all methods and modalities, as can be seen from Table 2. However, attention methods exhibit a substantially higher SD between faithfulness metrics compared to attribution methods (see Table 2), rendering the faithfulness results for attention methods more uncertain. Considering our concerns about the complexity metrics as well as the high SD between faithfulness metrics, we would subsequently advocate only for prioritizing attention methods over attribution methods if robustness is the most desired criterion.

**What are prominent trends for individual XAI methods?** The comprehensive evaluation further revealed distinct findings (1-4) relevant to selecting individual methods. **(1)** The most reliable performing method in terms of faithfulness and robustness is EG. We would recommend EG as an initial approach in various situations due to its weak dependence on hyperparameters and input modalities, as well as the always average SD between metrics.. This is especially important for data with non-triviality to select baseline values. **(2)** We observe a high variance in performance between methods that rely heavily on the gradient (VG, IxG, GB), with only the raw gradient VG being robust but not faithful, which can be

**a.** Correlation between XAI Methods ($PCC_{F_{ij}}$)

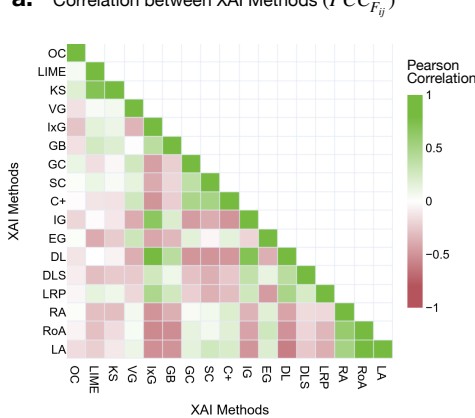

**b.** Attribution and Attention per Evaluation Criterion ($\bar{R}_{\bar{C},F=\{F_A,F_T\}}$)

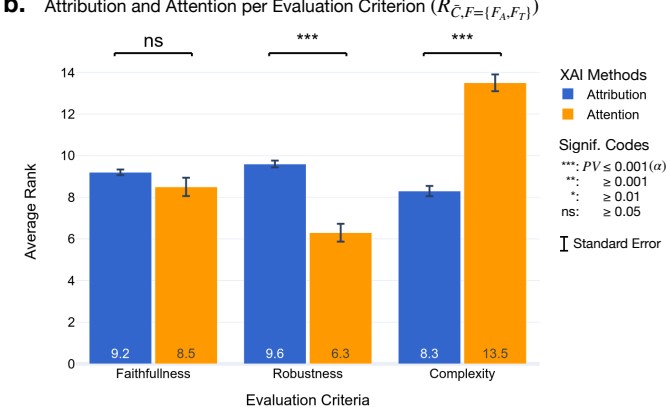

Figure 6: **a.** Correlation in ranking between XAI methods (see Equation 8). **b.** Average ranks of attribution (across all architectures, but across Transformers-only is very alike) and attention (Transformers-only) methods ($\bar{R}_{\bar{C},F=\{F_A,F_T\}}$). The standard error of the mean is larger for attention methods.

attributed to gradient shattering (Balduzzi et al., 2017). We would therefore advise against these methods or advance them through VarGrad or SmoothGrad. **(3)** Relevance-filtered attention (LA) in the majority of cases scores better than non-filtered raw attention. However, LA's SD between faithfulness metrics is unusually high (5.15) for a comparable good average rank of 3. When analyzing the metric disagreement in detail, we observe that the IROF and Faithfulness Estimate metrics rank the method as the worst compared to all other metrics which rank the method in the upper quarter (see Appendix J for a more detail analysis). Since only two metrics are clear outliers we would still prefer LA compared to non-filtered raw attention (which compared to attribution methods also has a high SD). Also, LA allows to visualize input features that attribute to a specific outcome and are not only detected by the model in general, making it much more versatile. **(4)** For some methods (EG, LRP, LA) we observe a trade-off between faithfulness and complexity. For LRP, its tendency to attribute to a very small set of input features likely explains this observation: Faithfulness is low due to the absence of important input features in the set, robustness is low as the relative change in this set can occur fast, but complexity, as evaluated in our metrics, is also low due to the small set size. We would argue that, theoretically, there can be a trade-off between faithfulness and complexity. However, we advise against overinterpreting these findings in light of our reservations regarding the complexity metrics and the subjectiveness of such a trade-off.

## 4.1 Main insights & takeaways

Our benchmark study presents key findings to bridge research gaps and inconsistencies. More specific findings are obtainable from our full ranking tables and the LATEC dataset. The most critical outcomes of our study can be condensed into the following main insights for evaluation and practical takeaways for XAI application:

**Insights for XAI evaluation**
1. For the same XAI method, there is frequently a risk of conflicting metrics, leading to unreliable rankings. However possible disagreement between metrics can provide essential insights about uncertainty in XAI evaluation.
2. Variations in ranking among metrics can be attributed to similarities in their mathematical mechanisms, designs that address limitations of other metrics, or the alignment between evaluation metrics and explanation methods. Selection of *only* metric pairs, or method and metric pairs that fall into this category, should be avoided.
3. Results in complexity seem counterintuitive, indicating that the evaluation objective of complexity metrics in computer vision does not always has to match the perception of low complexity.

**Takeaways for XAI application**

1. No XAI method ranks consistently high on all evaluation criteria. Curiously, EG performs most reliably, even if it was never examined in any relevant related study before.
2. Attention methods are mainly more robust compared to attribution methods, but have substantially larger SD between faithfulness metrics. Relevance-filtered attention (LA) consistently scores higher than non-filtered raw attention.
3. Rankings of XAI methods typically generalize well over datasets given not too distinct size and characteristics but can highly depend on modality (especially for CAM and linear surrogate methods).

## 5 Comparison with related work

As we previously stated, the significance of current research is limited due to small and varying subsets in XAI methods, metrics, and criteria used for evaluation, which consequently lead to contradicting outcomes. Due to the comprehensiveness and the more robust results of our study through the reporting of uncertainty measurements, we argue that the validity is significantly increased compared to previous work and that our findings offer a more definitive resolution to the inconsistencies observed in earlier, smaller-scale investigations. The difference in scale is particularly evident from Table 1, which presents a summary of 17 related relevant studies (all on image data as there are none for volume or point cloud), with an indication of what criterion is evaluated and how XAI methods performed relative to other methods or baselines analyzed in the study, based on our assessment. We present our results for the imaging modality indicated at the bottom, which do not necessarily transfer to other modalities as Table 1 shows. While some "evergreen" XAI methods, i.e., VG, IxG, GB, and IG, stand out, Table 1 visualizes how sparse the field of XAI evaluation is, especially for attention methods. Surprisingly, our best-evaluated method in terms of faithfulness and robustness, EG, is not evaluated in any related study.

Back-referencing to Table 1, we observe in several cases similar results to other studies on image data: low faithfulness of GB, IxG (as both methods perform partial input recovery), LIME or LRP by Chefer et al. (2021). High faithfulness of IG by Arras et al. (2022) and Hesse et al. (2023) (depends highly on the selected baseline) and LA (only two studies including attention methods). Regarding the conflicting outcomes reported for GC, our results show average faithfulness but high robustness on image data (but can depend on the underlying model, as our work suggests). On the contrary, our results contradict the findings on high faithfulness of LRP by Arras et al. (2022), low faithfulness of IG, and high robustness of KS (KS studies use lower dimensional image data). However, these results can differ between modalities, as GC, for example, obtains very low scores in faithfulness and robustness on volume data. No related studies that examine both attention and attribution methods address the notably higher SD observed in attention methods compared to attribution methods when evaluating faithfulness.

Most work in complexity and human understandability is qualitative and only quantitative work includes papers that present a metric. We consider the high fluctuation between quantitative and especially qualitative complexity evaluation outcomes by Singh et al. (2021) as further support for our hypothesis that there is a gap between the aim of the metrics and human conception of low complexity. As also the human conception of low complexity can be very subjective, we strongly recommend the development of either new metrics or falling back to robust qualitative user studies, even if they are more difficult and resource-consuming.

## 6 Conclusion and discussion

Although our benchmark is one of the most comprehensive in the field, we restrict us to the modalities with the, in our opinion, most unique and not overlapping characteristics, ignoring e.g. video or language data, as this would also introduce modality-specific XAI methods and metrics. Given the significant discrepancies observed in computer vision modalities, we anticipate potentially larger variations in evaluations across even less related modalities. Further, we do not include unconventional post-hoc XAI methods such as symbolic representations or metamodels due to their low rate of adoption in practice. Our approach selectively incorporates the most informative criteria and deliberately excludes rather niche criteria such as localization or axiomatic properties as they either require ground-truth bounding boxes or can not be applied to all XAI methods. While confirmation of a method's adherence to its axioms is valuable for academic inquiry into

mathematical behaviors, it offers limited practical guidance for users selecting a method. The benchmark focuses on the comparison between methods, not on the evaluation of whether a method is faithful or robust in general, thus ignoring e.g. synthetic baselines. Our findings demonstrate high generalizability, yet it's crucial to contextualize them within the benchmark, considering possible variance introduced by real-world and large-scale scenarios.

Our results demonstrate vividly the need for rethinking the application of evaluation metrics and the risks of inconsistent benchmarking for practitioners and researchers. As a solution, we offer practitioners concise, practical takeaways for applying and selecting XAI methods derived from our large-scale benchmark. This includes the most all-encompassing answer to *"What XAI method should I (not) use for my problem?"* to date, based on the extensive evidence in our provided results tables and the LATEC dataset. Through these detailed results tables, practitioners can evaluate which method is most applicable or should be avoided in their specific setting concerning different criteria. After method selection, practitioners can additionally leverage LATEC to generate diverse and large-scale sets of saliency maps including different computer vision modalities. For researchers, we advance current mechanisms of evaluation, address the risk of conflicting metrics, and introduce LATEC as a platform for standardized benchmarking of methods and metrics in XAI. LATEC offers researchers the opportunity to explore and answer numerous critical questions regarding the trustworthiness of XAI, thereby playing a pivotal role in the advancement of the field. We hope that our systematic approach to addressing significant shortcomings in existing research will provide a clearer and more reliable path through the intricate maze of methods and metrics in XAI.

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

# Appendix

## A Model performance and hyperparameter

### A.1 Test set performance

**a.** Testset Performance on Image Modality

**Model Performance Metric**

| Dataset: | Model Architecture: | Accuracy | Precision | Recall | F1 | AUROC |
|---|---|---|---|---|---|---|
| OCT (MC: 4) | ResNet 50 | 0.999 | 0.999 | 0.999 | 0.999 | 1.0 |
| | EfficientNet b0 | 0.9969 | 0.9969 | 0.9969 | 0.9969 | 1.0 |
| | DeiT ViT | 0.999 | 0.999 | 0.999 | 0.999 | 1.0 |
| R45 (MC: 45) | ResNet 50 | 0.9535 | 0.9536 | 0.9538 | 0.9535 | 0.9995 |
| | EfficientNet b0 | 0.9554 | 0.9554 | 0.9549 | 0.9549 | 0.9995 |
| | DeiT ViT | 0.9568 | 0.957 | 0.9568 | 0.9567 | 0.9995 |

**b.** Testset Performance on Volume Modality

**Model Performance Metric**

| Dataset: | Model Architecture: | Accuracy | Precision | Recall | F1 | AUROC |
|---|---|---|---|---|---|---|
| AMN (BC) | 3D ResNet 18 | 0.8003 | 0.8013 | 0.7987 | 0.8 | 0.8699 |
| | EfficientNet3D b0 | 0.8003 | 0.7954 | 0.8087 | 0.802 | 0.8647 |
| | Simple3DFormer | 0.7936 | 0.7907 | 0.7907 | 0.7907 | 0.8728 |
| OMN (MC: 11) | 3D ResNet 18 | 0.9115 | 0.9248 | 0.9248 | 0.9226 | 0.9953 |
| | EfficientNet3D b0 | 0.8754 | 0.8924 | 0.8936 | 0.8914 | 0.9893 |
| | Simple3DFormer | 0.8131 | 0.8463 | 0.8381 | 0.84 | 0.9815 |
| VMN (BC) | 3D ResNet 18 | 0.9359 | 0.937 | 0.9346 | 0.9358 | 0.98 |
| | EfficientNet3D b0 | 0.9162 | 0.9162 | 0.9162 | 0.9162 | 0.9229 |
| | Simple3DFormer | 0.8861 | 0.8871 | 0.8848 | 0.886 | 0.9394 |

**c.** Testset Performance on Point Cloud Modality

**Model Performance Metric**

| Dataset: | Model Architecture: | Accuracy | Precision | Recall | F1 | AUROC |
|---|---|---|---|---|---|---|
| CMA (MC: 12) | PointNet | 0.9852 | 0.9743 | 0.9876 | 0.98 | 0.998 |
| | DGCNN | 0.9535 | 0.9373 | 0.9498 | 0.9423 | 0.9989 |
| | PC Transformer | 0.9751 | 0.9645 | 0.9688 | 0.9662 | 0.9996 |
| M40 (MC: 40) | PointNet | 0.8914 | 0.8374 | 0.8564 | 0.8438 | 0.9958 |
| | DGCNN | 0.9177 | 0.8844 | 0.891 | 0.8864 | 0.9973 |
| | PC Transformer | 0.9149 | 0.8779 | 0.8842 | 0.8796 | 0.9969 |
| SHN (MC: 16) | PointNet | 0.9878 | 0.9673 | 0.9689 | 0.9668 | 0.9991 |
| | DGCNN | 0.9903 | 0.966 | 0.9847 | 0.9745 | 0.9995 |
| | PC Transformer | 0.9896 | 0.9642 | 0.9819 | 0.9716 | 0.9997 |

**MC #**: Multi-Class (# Classes), **BC**: Binary-Class

Table 3: **a., b. & c.** Test set performance measured with the metrics: accuracy, precision, recall, F1, and area under the receiver operating characteristic (AUROC) curve, for each modality. In the case of IMN we use pretrained weights for the Transformer architecture from Huggingface[1] and the CNN architectures from TorchHub[2,3].

[1] `https://huggingface.co/facebook/deit-small-patch16-224`

[2] `https://pytorch.org/vision/stable/models/generated/torchvision.models.resnet50.html`

[3] `https://pytorch.org/vision/stable/models/generated/torchvision.models.efficientnet_b0.html`

Architectures were chosen based on their popularity and, to a limited extent, comparability between modalities, e.g. ResNet-50 and 3D ResNet-18 which both emerge from the same family of ResNet architectures. While 3D volume architectures could also be applied to point cloud data, we choose point cloud specific architectures for the modality.

## A.2 Hyperparameter

We tuned all hyperparameters on either the declared validation set or sampled a validation set based on 20% of the train set. The tuning was performed via grid search for each model. The primary metric for hyperparameter tuning was the F1 score.

Hyperparameter for Image Modality

| Model Architecture: | Hyperparameter: | Utilized Computer Vision Datasets | |
|---|---|---|---|
| | | OCT | R45 |
| ResNet 50 | Batch size | 128 | 128 |
| | Max Epochs | 8 | 60 |
| | Learning rate (LR) | 0.0001 | 0.0001 |
| | Optimizer | Madgrad | Madgrad |
| | LR Scheduler | Cosine Annealing | Cosine Annealing |
| | Weight Decay | 0 | 0 |
| | Momentum | 0.9 | 0.9 |
| | Augmentations | Train:
Resize (256,256)
RandomCrop (224,224)
RandomAffine (shear=0.2, degrees=5)
RandomHorizontalFlip
Grayscale (channels=3)

Test:
Resize (256,256)
CenterCrop (224,224)
Grayscale (channels=3) | Train:
Resize (256,256)
RandomCrop (224,224)
RandomHorizontalFlip
RandAugment
Normalize (mean=(0.485, 0.456, 0. 406), std=(0.229, 0.224, 0.225))

Test:
Resize (256,256)
Normalize (mean=(0.485, 0.456, 0. 406), std=(0.229, 0.224, 0.225)) |
| | Sampling | Weighted Random Sampling | None |
| EfficientNet b0 | Batch size | 128 | 128 |
| | Max Epochs | 5 | 15 |
| | Learning rate (LR) | 0.0001 | 0.001 |
| | Optimizer | Madgrad | Madgrad |
| | LR Scheduler | Cosine Annealing | Cosine Annealing |
| | Weight Decay | 0 | 0 |
| | Momentum | 0.9 | 0.9 |
| | Augmentations | Train:
Resize (256,256)
RandomCrop (224,224)
RandomAffine (shear=0.2, degrees=5)
RandomHorizontalFlip
Grayscale (channels=3)

Test:
Resize (256,256)
CenterCrop (224,224)
Grayscale (channels=3) | Train:
Resize (256,256)
RandomCrop (224,224)
RandomHorizontalFlip
RandAugment
Normalize (mean=(0.485, 0.456, 0. 406), std=(0.229, 0.224, 0.225))

Test:
Resize (256,256)
Normalize (mean=(0.485, 0.456, 0. 406), std=(0.229, 0.224, 0.225)) |
| | Sampling | Weighted Random Sampling | None |
| DeIT ViT | Batch size | 128 | 128 |
| | Max Epochs | 6 | 60 |
| | Learning rate (LR) | 0.0001 | 0.0001 |
| | Optimizer | Madgrad | Madgrad |
| | LR Scheduler | Cosine Annealing | Cosine Annealing |
| | Weight Decay | 0 | 0 |
| | Momentum | 0.9 | 0.9 |
| | Augmentations | Train:
Resize (256,256)
RandomCrop (224,224)
RandomAffine (shear=0.2, degrees=5)
RandomHorizontalFlip
Grayscale (channels=3)

Test:
Resize (256,256)
CenterCrop (224,224)
Grayscale (channels=3) | Train:
Resize (256,256)
RandomCrop (224,224)
RandomHorizontalFlip
RandAugment
Normalize (mean=(0.485, 0.456, 0. 406), std=(0.229, 0.224, 0.225))

Test:
Resize (256,256)
Normalize (mean=(0.485, 0.456, 0. 406), std=(0.229, 0.224, 0.225)) |
| | Sampling | Weighted Random Sampling | None |

Table 4: Hyperparameter for all three architectures and CV datasets, excluding IMN as we load pretrained weights.

Hyperparameter for Volume Modality

**Utilized Computer Vision Datasets**

| Model Architecture: | Hyperparameter: | AMN | OMN | VMN |
|---|---|---|---|---|
| **3D ResNet18** | Batch size | 32 | 32 | 32 |
| | Max Epochs | 100 | 100 | 100 |
| | Learning rate (LR) | 0.001 | 0.001 | 0.001 |
| | Optimizer | SGD | Adam | Adam |
| | LR Scheduler | Cosine Annealing | Cosine Annealing | Cosine Annealing |
| | Weight Decay | 0 | 0 | 0 |
| | Momentum | 0.9 | 0 | 0 |
| | Augmentations | Train: RandomBrightness($U(0,1)$)  Test: FixedBrightness(0.5) | None | Train: RandomBrightness($U(0,1)$)  Test: FixedBrightness(0.5) |
| | Sampling | Weighted Random Sampling | None | Weighted Random Sampling |
| **3D EfficientNet b0** | Batch size | 32 | 32 | 64 |
| | Max Epochs | 100 | 100 | 100 |
| | Learning rate (LR) | 0.001 | 0.001 | 0.001 |
| | Optimizer | SGD | AdamW | Adam |
| | LR Scheduler | Cosine Annealing | Cosine Annealing | Cosine Annealing |
| | Weight Decay | 0.0005 | 0.0005 | 0 |
| | Momentum | 0.9 | 0 | 0 |
| | Augmentations | Train: RandomBrightness($U(0,1)$)  Test: FixedBrightness(0.5) | None | Train: RandomBrightness($U(0,1)$)  Test: FixedBrightness(0.5) |
| | Sampling | Weighted Random Sampling | None | Weighted Random Sampling |
| **Simple3DFormer** | Batch size | 32 | 32 | 64 |
| | Max Epochs | 150 | 100 | 100 |
| | Learning rate (LR) | 0.001 | 0.000001 | 0.001 |
| | Optimizer | SGD | Madgrad | Adam |
| | LR Scheduler | Cosine Annealing | Cosine Annealing | Cosine Annealing |
| | Weight Decay | 0.0005 | 0 | 0 |
| | Momentum | 0.9 | 0.9 | 0 |
| | Augmentations | Train: RandomBrightness($U(0,1)$)  Test: FixedBrightness(0.5) | None | Train: RandomBrightness($U(0,1)$)  Test: FixedBrightness(0.5) |
| | Sampling | Weighted Random Sampling | None | Weighted Random Sampling |

Table 5: Hyperparameters for all three architectures and CV datasets.

Hyperparameter for Point Cloud Modality

| Model Architecture: | Hyperparameter: | Utilized Computer Vision Datasets | | |
|---|---|---|---|---|
| | | CMA | M40 | SHN |
| PointNet | Batch size | 32 | 24 | 32 |
| | Max Epochs | 100 | 200 | 200 |
| | Learning rate (LR) | 0.001 | 0.001 | 0.001 |
| | Optimizer | AdamW | AdamW | AdamW |
| | LR Scheduler | Cosine Annealing | Cosine Annealing | Cosine Annealing |
| | Weight Decay | 0.0001 | 0.0001 | 0.0001 |
| | Momentum | 0 | 0 | 0 |
| | Augmentations | Pretransforms: NormalizeScale

Train: SamplePoints (1024) RandomScale (0.67,1.5) RandomRotate (degrees=15) RandomJitter (0.02)

Test: SamplePoints (1024) | Pretransforms: NormalizeScale

Train: SamplePoints (1024) RandomScale (0.67,1.5) RandomJitter (0.02)

Test: SamplePoints (1024) | Pretransforms: NormalizeScale

Train: RandomScale (0.67,1.5) RandomJitter (0.01) RandomRotate(degress=15, axis = (0,1,2))

Test: None |
| | Sampling | None | None | None |
| DGCNN | Batch size | 32 | 32 | 32 |
| | Max Epochs | 100 | 250 | 200 |
| | Learning rate (LR) | 0.001 | 0.001 | 0.001 |
| | Optimizer | AdamW | AdamW | AdamW |
| | LR Scheduler | Cosine Annealing | Cosine Annealing | Cosine Annealing |
| | Weight Decay | 0.0001 | 0.0001 | 0.0001 |
| | Momentum | 0 | 0 | 0 |
| | Augmentations | Pretransforms: NormalizeScale

Train: SamplePoints (1024) RandomScale (0.67,1.5) RandomRotate (degrees=15) RandomJitter (0.02)

Test: SamplePoints (1024) | Pretransforms: NormalizeScale

Train: SamplePoints (1024) RandomScale (0.67,1.5) RandomJitter (0.02)

Test: SamplePoints (1024) | Pretransforms: NormalizeScale

Train: RandomScale (0.67,1.5) RandomJitter (0.01) RandomRotate(degress=15, axis = (0,1,2))

Test: None |
| | Sampling | None | None | None |
| PC Transformer | Batch size | 32 | 32 | 32 |
| | Max Epochs | 150 | 250 | 200 |
| | Learning rate (LR) | 0.01 | 0.01 | 0.01 |
| | Optimizer | SGD | SGD | SGD |
| | LR Scheduler | Cosine Annealing | Cosine Annealing | Cosine Annealing |
| | Weight Decay | 0.0005 | 0.0005 | 0.0005 |
| | Momentum | 0.9 | 0.9 | 0.9 |
| | Augmentations | Pretransforms: NormalizeScale

Train: SamplePoints (1024) RandomScale (0.67,1.5) RandomRotate (degrees=15) RandomJitter (0.02)

Test: SamplePoints (1024) | Pretransforms: NormalizeScale

Train: SamplePoints (1024) RandomScale (0.67,1.5) RandomJitter (0.02)

Test: SamplePoints (1024) | Pretransforms: NormalizeScale

Train: RandomScale (0.67,1.5) RandomJitter (0.01) RandomRotate(degress=15, axis = (0,1,2))

Test: None |
| | Sampling | None | None | None |

Table 6: Hyperparameters for all three architectures and CV datasets.

## B   The LATEC dataset: Reference data for standardized evaluation

The resulting data of the three stages, which comprise the LATEC dataset, include pretrained model weights (excluding IMN), saliency maps, and evaluation scores. Thanks to the LATEC dataset, future experiments can start at a certain stage and use the results from the previous stage without recomputing everything again, e.g. when testing out a new evaluation metric on the existing saliency maps, preserving comparability. For the LATEC dataset, we compute per dataset saliency maps for the entire test set or 1000 observations depending on which size is smaller (on the validation set if the test set is unavailable), from which we sample 50 observations to compute evaluation scores for all 7,560 combinations. In total, the LATEC dataset consists of 326,790 saliency maps and 378,000 evaluation scores. As for such large datasets, the size can go into the hundreds of gigabytes. To save disk space, saliency maps could be cast from 64-bit precision to 32 or even 16-bit. We would, however, strongly advise against this, as even casting to 32-bit precision introduced numerical instability in our experiments due to the rounding of attribution and attention values, resulting in all-zero saliency maps and *nan* or *inf* evaluation scores. Further, as ranking lengths between CNN and Transformer architectures differ (attention methods only for Transformer architectures), we recompute rankings in the subsequent study, which aggregate over all three architectures by first combining the normalized evaluation

Figure 7: Illustrative saliency maps for all three modalities. The upper row shows three attributions, respectively, and the lower row, three attention-based methods. We observe how all XAI methods highlight the runway in the image and the vessel for the volume modality but with different granularity and focus. For the point cloud plane, explanations are less understandable, with attribution methods highlighting single points at the front tip, rudder, or wing tips.

scores per model architecture and then computing the ranking, preserving equal length between rankings (see Appendix G).

To ensure a standardized setting with fair comparability between XAI methods over all possible experiment set-ups and aggregation levels, we take precautions regarding e.g. different types of feature attributions or the conversion of all metrics to single scores (see Appendix F for all detailed procedures). LRP requires non-negative activation outputs Montavon et al. (2019), leading us to a replacement of such activation functions (i.e. GeLU, leakyReLU) in CNN models, but we keep them for Transformer models, as they are central to the architecture and therefore also to our benchmark, and apply the $0^+$-rule instead.

## C  XAI methods overview and parameters

### C.1  Overview

#### C.1.1  Attribution Methods

**Occlusion [OC]** (Zeiler & Fergus, 2013)    Systematically obscures different parts of the input data and observes the resulting impact on the output, to determine which parts of the data are most important for the model's predictions.

**LIME** [LIME] (Ribeiro et al., 2016)    Creates an interpretable model around the prediction of a complex model to explain individual predictions locally (patch-based in our case), using perturbations of the input data and observing the corresponding changes in the output.

**Kernel SHAP [KS]** (Lundberg & Lee, 2017)    Using a weighted linear regression model as the local surrogate and selecting a suitable weighting kernel, the regression coefficients from the LIME surrogate can estimate the SHAP values.

**Vanilla Gradient [VG]** (Simonyan et al., 2013)    The raw input gradients of the model.

**Input x Gradient [IxG]** (Shrikumar et al., 2017)    Multiples the input features by their corresponding gradients with respect to the model's output.

**Guided Backprob [GB]** (Springenberg et al., 2015)    Modifies the standard backpropagation process to only propagate positive gradients for positive inputs through the network, thereby creating visualizations that highlight the features that strongly activate certain neurons in relation to the target output.

**GradCAM [GC]** (Selvaraju et al., 2017)    Uses the gradients of the target class flowing into the final convolutional layer to produce a coarse localization map by, highlighting the important regions in the image by up-scaling the map.

**ScoreCAM [SC]** (Wang et al., 2020)   Eliminates the need for gradient information by determining the importance of each activation map based on its forward pass score for the target class, producing the final output through a weighted sum of these activation maps.

**GradCAM++ [C+]** (Chattopadhay et al., 2018)   Generates a visual explanation for a given class label by employing a weighted sum of the positive partial derivatives from the final convolutional layer's feature maps, using them as weights with respect to the class score.

**Integrated Gradients [IG]** (Sundararajan et al., 2017)   Explains model predictions by attributing the prediction to the input features, calculating the path integral of the gradients along the straight-line path from a baseline input to the actual input.

**Expected Gradients [EG]** (Erion et al., 2020)   Also called Gradient SHAP. Avoids the selection of a baseline value compared to IG, by leveraging a probabilistic baseline computed over a sample of observations.

**DeepLIFT [DL]** (Shrikumar et al., 2017)   Assigns contribution scores to each input feature based on the difference between the feature's activation and a reference activation, effectively measuring the feature's impact on the output compared to a baseline.

**DeepLIFT SHAP [DLS]** (Lundberg & Lee, 2017)   Combines the DeepLIFT method with Shapley values to assign importance scores to input features by computing their contributions to the output relative to a reference input, while ensuring consistency with Shapley values.

**Layer-Wise Relevance Propagation [LRP]** (Binder et al., 2016)   Explains neural network decisions by backpropagating the output prediction through the layers, redistributing relevance scores to the input features to visualize their contribution to the final decision. We use the $\epsilon$-,$\gamma$- and $0^+$-rules depending on the model architecture for relevance backpropagation.

### C.1.2   Attention Methods

**Raw Attention [RA]** (Dosovitskiy et al., 2021)   Rearranged and up-scaled attention values of the last attention head.

**Rollout Attention [RoA]** (Abnar & Zuidema, 2020)   Averages attention weights of multiple heads to trace the contribution of each part of the input data through the network.

**LRP Attention [LA]** (Chefer et al., 2021)   Assigns local relevance scores to attention weights based on the Deep Taylor Decomposition principle and propagates these relevancy scores through the model.

### C.2   Parameters

| XAI Method: | OC | | | | LIME | | | KS | | | CAM (all) | SC | IG | | EG | | DL | | LRP | | | RA |
|---|---|---|---|---|---|---|---|---|---|---|---|---|---|---|---|---|---|---|---|---|---|---|
| Parameter: | strides | sliding_window_shapes | baseline | perturbations_per_eval | alpha | n_samples | perturbations_per_eval | baseline | n_samples | perturbations_per_eval | layer | batch_size | baseline | n_steps | n_samples | std | eps | baseline | rule | eps | gamma | layer |
| **Image** | 25 | (50, 50) | 0 | 1 | 1,0 | 10 | 5 | 0 | 10 | 5 | ResNet50,layer4[-1] EfficientNetbo,features[-1] ViT,blocks[-1],norm1 | 32 | 0 | 30 | 40 | 0,001 | 1e-9 | 0 | ε & γ-rule / 0+-rule | 0,0001 | 0,25 | ViT,blocks[-1],attn |
| **Volume** | 4 | (7, 7, 7) | 0 | 1 | 1,0 | 10 | 5 | 0 | 10 | 5 | 3DEfficientNetbo,blocks[-13] 3DResNet18,layer3 S3DF,blocks[-1],norm1 | 64 | 0 | 30 | 40 | 0,001 | 1e-9 | 0 | ε & γ-rule / 0+-rule | 0,0001 | 0,25 | S3DF[-1],attn |
| **Point Cloud** | 1 | (3,1) | 0 | 5 | 4,0 | 10 | 5 | 0 | 10 | 5 | PointNet,transform,bn1 DGCNN,conv5 PCT,sa4,after_norm | 16 | 0 | 30 | 16 | 0,001 | 1e-9 | 0 | ε & γ-rule / 0+-rule | 0,00001 | 0,25 | PCT,sa4,attn |

Table 7: Parameters for each XAI method and modality.

The parameters for each XAI method are derived for each modality via qualitative evaluation which we deem the most realistic scenario. We tuned the XAI methods on five observations per dataset and modality, which we argue is a fair trade-off between fitting the methods to the dataset but not overfitting them to bias the evaluation. We did not tune the parameters per dataset, as the parameters transfer very well between datasets and only needed minimal adjustments.

# D  Evaluation metrics overview and parameters

## D.1  Overview

### D.1.1  Faithfulness

**Faithfulness Correlation** (Bhatt et al., 2020)    Gauges an explanation's fidelity to model behavior. It measures the linear correlation between predicted logits of modified test points and the average explanation for selected features, returning a score between -1 and 1. For each test, selected features are replaced with baseline values, and Pearson's correlation coefficient is determined, averaging results over multiple tests.

**Faithfulness Estimate** (Alvarez Melis & Jaakkola, 2018)    Evaluates the accuracy of estimated feature relevances by using a proxy for the "true" influence of features, as the actual influence is often unavailable. This is done by observing how the model's prediction changes when certain features are removed or obscured. Specifically, for probabilistic classification models, the metric looks at how the probability of the predicted class drops when features are removed. This drop is then compared to the interpreter's prediction of that feature's relevance. The metric also computes correlations between these probability drops and relevance scores across various data points.

**Monotonicity Correlation** (Nguyen & Martínez, 2020)    Evaluates the correlation between the absolute values of attributions and the uncertainty in probability estimation using Spearman's coefficient. If attributions are not monotonic the authors argue that they are not providing the correct importance of the features.

**Pixel Flipping** (Bach et al., 2015)    The core concept involves flipping pixels with very high, very low, or near-zero attribution scores. The effect of these changes is then assessed on the prediction scores, with the average prediction being determined.

**Region Perturbation** (Samek et al., 2017)    A step-by-step method where the class representation in the image, as determined by a function, diminishes as we gradually eliminate details from an image. This process, known as region perturbation, occurs at designated locations. Finally, the effect on the average prediction is calculated.

**Insertion** (Petsiuk et al., 2018)    Gradually inserts features into a baseline input, which is a strongly blurred version of the image, to not create OOD examples. During this process, the change in prediction is measured and the correlation with the respective attribution value is calculated.

**Deletion** (Petsiuk et al., 2018)    Deletes input features one at a time by replacing them with a baseline value based on their attribution score. During this process, the change in prediction is measured and the correlation with the respective attribution value is calculated.

**Iterative Removal of Features (IROF)** (Rieger & Hansen, 2020)    The metric calculates the area under the curve for each class based on the sorted average importance of feature segments (superpixels). As these segments are progressively removed and prediction scores gathered, the results are averaged across multiple samples.

**Remove and Debias (ROAD)** (Rong et al., 2022)    Evaluates the model's accuracy on a sample set during each phase of an iterative process where the k most attributed features are removed. To eliminate bias, in every step, the k most significant pixels, by the most relevant first order, are substituted with noise-infused linear imputations.

**Sufficiency** (Dasgupta et al., 2022)    Assesses the likelihood that the prediction label for a specific observation matches the prediction labels of other observations which have similar saliency maps.

### D.1.2  Robustness

**Local Lipschitz Estimate** (Alvarez Melis & Jaakkola, 2018)    Lipschitz continuity in calculus is a concept that measures the relative changes in a function's output concerning its input. While the traditional definition of Lipschitz continuity is global, focusing on the largest relative deviations across the entire input space,

this global perspective isn't always meaningful in XAI. This is because expecting consistent explanations for vastly different inputs isn't realistic. Instead, a more localized approach, focusing on stability for neighboring inputs, is preferred, resulting in a point-wise, neighborhood-based local Lipschitz continuity metric.

**Max Sensitivity** (Yeh et al., 2019)    Measures the largest shift in the explanation when the input is slightly altered. It specifically evaluates the utmost sensitivity of a saliency map by taking multiple samples from a defined L-infinity ball subspace with a set input neighborhood radius, using Monte Carlo sampling for approximation.

**Continuity** (Montavon et al., 2018)    Evaluates, that if two observations are nearly equivalent, then the explanations of their predictions should also be nearly equivalent. It then measures the strongest variation of the explanation in the input domain.

**Relative Input/Output/Representation Stability** (Agarwal et al., 2022)    All metrics leverage model information to evaluate the stability of a saliency map with respect to the change in the either, input data, intermediate representations, and output logits of the underlying prediction model.

**Infidelity** (Yeh et al., 2019)    Calculates the expected mean-squared error (MSE) between the saliency map multiplied by a random variable input perturbation and the differences between the model at its input and perturbed input. We use Infidelity with the noisy perturbation baseline.

### D.1.3   Complexity

**Sparseness** (Chalasani et al., 2020)    Measures the Gini Index on the vector of absolute saliency map values. The assessment ensures that features genuinely influencing the output have substantial contributions, while insignificant or only slightly relevant features should have minimal contributions.

**Complexity** (Bhatt et al., 2020)    Determines the entropy of the normalized saliency map.

**Effective Complexity** (Nguyen & Martínez, 2020)    Evaluates the number of absolute saliency map values that surpass a threshold. Values above this threshold suggest the features are significant, while those below indicate they are not.

## D.2   Parameters

We tuned the parameters of the evaluation metrics per dataset based on the distribution of their scores from Appendix O. We applied the suggested parameters from Hedström et al. (2022) or the respective papers. If the resulting score distributions were collapsed, almost uniform, or too indistinguishable between the XAI methods, we tuned the respective parameters. This step was completed prior to the ranking analysis, and no adjustments were made to the metrics once the ranking phase commenced.

| Evaluation Metric: | Parameter: | Image | | | Voxel | | | Point Cloud | | |
|---|---|---|---|---|---|---|---|---|---|---|
| | | IMN | OCT | R45 | AMN | OMN | VMN | CMA | M40 | SHN |
| Faithfulness Correlation | nr_runs | 100 | 100 | 100 | 100 | 100 | 100 | 100 | 100 | 100 |
| | subset_size | 224 | 224 | 224 | 56 | 56 | 56 | 32 | 32 | 32 |
| | perturb_baseline | black | black | black | black | black | black | center | center | center |
| Faithfulness Estimate | features_in_step | 224 | 224 | 224 | 56 | 56 | 56 | 32 | 32 | 32 |
| | perturb_baseline | black | black | black | black | black | black | center | center | center |
| Monotonicity Correlation | nr_samples | 10 | 10 | 10 | 10 | 10 | 10 | 10 | 10 | 10 |
| | features_in_step | 3136 | 3136 | 3136 | 392 | 392 | 392 | 256 | 256 | 256 |
| | perturb_baseline | uniform | uniform | uniform | uniform | uniform | uniform | uniform | uniform | uniform |
| Pixel Flipping | features_in_step | 224 | 224 | 224 | 56 | 56 | 56 | 32 | 32 | 32 |
| | perturb_baseline | black | black | black | black | black | black | center | center | center |
| Region Perturbation | patch_size | 14 | 14 | 18 | 4 | 4 | 4 | 3 | 3 | 3 |
| | regions_evaluation | 10 | 10 | 20 | 20 | 20 | 20 | 32 | 32 | 32 |
| | perturb_baseline | uniform | uniform | uniform | uniform | uniform | uniform | uniform | uniform | uniform |
| Insertion | pixel_batch_size | 50 | 50 | 50 | 50 | 50 | 50 | 50 | 50 | 50 |
| | sigma | 5.0 | 120.0 | 40.0 | 2.5 | 2.5 | 2.5 | 0.05 | 0.1 | 0.05 |
| | kernel_size | 15 | 39 | 19 | 1 | 1 | 1 | 1 | 1 | 1 |
| Deletion | pixel_batch_size | 50 | 50 | 50 | 50 | 50 | 50 | 50 | 50 | 50 |
| IROF | segmentation | Slic | Slic | Slic | 3D Slic | 3D Slic | 3D Slic | KMeans | KMeans | KMeans |
| | perturb_baseline | mean | mean | mean | black | black | black | center | center | center |
| ROAD | noise | 0.1 | 0.1 | 0.1 | 4.0 | 2.5 | 50.0 | 0.02 | 0.15 | 0.3 |
| | percentages_max | 100 | 100 | 100 | 100 | 100 | 100 | 100 | 100 | 100 |
| Sufficiency | threshold | 0.9 | 0.6 | 0.6 | 0.02 | 0.75 | 0.0002 | 0.75 | 0.75 | 0.6 |
| Local Lipschitz Estimate | nr_samples | 5 | 5 | 5 | 10 | 10 | 10 | 5 | 5 | 5 |
| | perturb_std | 0.1 | 0.0002 | 0.1 | 0.2 | 0.2 | 0.2 | 0.1 | 0.1 | 0.1 |
| | perturb_mean | 0.0 | 0.0 | 0.0 | 0.0 | 0.0 | 0.0 | 0.0 | 0.0 | 0.0 |
| MaxSensitivity | nr_samples | 10 | 10 | 10 | 10 | 10 | 10 | 10 | 10 | 10 |
| | lower_bound | 0.2 | 0.2 | 0.2 | 0.2 | 0.2 | 0.2 | 0.2 | 0.2 | 0.2 |
| Continuity | patch_size | 56 | 56 | 56 | 7 | 7 | 7 | 3 | 3 | 3 |
| | nr_steps | 20 | 20 | 20 | 20 | 20 | 20 | 20 | 20 | 20 |
| | perturb_baseline | uniform | uniform | uniform | uniform | uniform | uniform | uniform | uniform | uniform |
| RIS | nr_samples | 10 | 10 | 10 | 10 | 10 | 10 | 10 | 10 | 10 |
| ROS | nr_samples | 10 | 10 | 10 | 10 | 10 | 10 | 10 | 10 | 10 |
| RRS | nr_samples | 10 | 10 | 10 | 10 | 10 | 10 | 10 | 10 | 10 |
| Infidelity | n_perturb_samples | 50 | 50 | 50 | 50 | 50 | 50 | 50 | 50 | 50 |
| Effective Complexity | eps | 0.01 | 0.01 | 0.01 | 0.001 | 0.001 | 0.001 | 0.001 | 0.001 | 0.001 |

Table 8: Parameters for all evaluation metrics on each CV dataset.

# E  Adapting current XAI methods and evaluation metrics for 3D data

While many XAI methods and evaluation metrics are independent of the input space dimensions, especially methods leveraging perturbations, interpolations for up- and down-scaling or segmentation are not. Our implementation builds upon the work from Kokhlikyan et al. (2020) and Hedström et al. (2023) for XAI methods and evaluation metrics for 1D and 2D images, and we extended it to 3D volume and point cloud data. Both modalities come with their own specifies, e.g. that local neighborhoods have to be defined via k-nearest neighbors (KNN) in point cloud data and not 2D or 3D patches as in image or volume data. For the XAI methods, we advanced e.g. OC, LI, and KS by the adoption of 3D patches, all three CAM methods with 3D interpolation, all attention-based methods with 3D and KNN-based interpolations, and LA with relevance backpropagation for the Simple3DFormer and PC Transformer architectures. As the adoption of the CAM methods for point cloud data and more complex architectures than PointNet is not trivial, we deem it out of scope for this paper and do not include them in our point cloud experiments. In the case of evaluation metrics, we adapted e.g. perturbation applying metrics to 3D patches or point-based perturbations, the superpixel segmentation in IROF by 3D Slic and KMeans clustering and padded x-axis transversal for the volume and point cloud data in Continuity. Additionally, we modified all methods and metrics to function with $(x, y, z)$ volume and $(n, 3)$ point cloud dimensions. All adaptations were tested for their coherency, and illustrative saliency maps can be observed in Figure 7. We refer to Appendix C and Appendix subsection D.2 for all implementation details.

## E.1  Adaption of XAI methods

In this section, we explain how we adapted XAI methods in our framework to seamlessly work with 3D modalities. We neglect the methods that did not need any adaption (besides e.g. unit tests etc.) as they work independently of the input dimensions. All XAI methods are adapted, such that they only return positive attribution.

**Occlusion**    For the 3D modalities we implemented a 3D kernel as the perturbation baseline for volumes and a 1x3 mask (one point) for the point clouds. The image and volume mask transverse with overlap and the point cloud mask without overlap over all dimensions of the input object.

**LIME & Kernel SHAP**    For both methods, we implemented feature masks for each modality, as training the linear surrogate models on the original input features is not informative and computationally very expensive. Each mask groups the input features to the same interpretable feature. We use predefined grids as feature masks, as superpixel computing algorithms are too computational and time-expensive, especially for 3D modalities and evaluation metrics that perturb the input space or refit the XAI method multiple times. For the image modality, we use a 16x16x3, for volume 7x7x7, and for point cloud 1x3 (one point) mask, which is distributed as a non-overlapping grid in all dimensions over the whole object. For point clouds we use ridge regression and for the other modalities lasso regression.

**GradCAM, ScoreCAM & GradCAM++**    For all CAM methods on volume data we adapted the gradient averaging and the subsequent weighting of the activations and used nearest-neighbor interpolation to upscale the weighted activations to 3D volumes. In the case of ScoreCAM we also use nearest neighbor up-sampling instead of bilinear up-sampling, to upscale the activations for weighting the output of the previous layer. To correctly reshape the upscaled images and volumes in the case of the Transformer architectures (taking the channels to the first dimension as for CNNs), we use two different reshape functions for images and volumes when the CAM methods are applied to Transformer architectures. Further, we use the absolute activation output, not the non-negative for Transformer architectures, as the leaky-ReLU/GeLU function output otherwise would sometimes be zero.

**LRP**    For CNNs, we assigned the $\epsilon$-rule to the linear or identity layers, the identity rule to all non-linear layers, and to all other layers (convolutions, pooling, batch normalization, etc.) the $\gamma$-rule. For Transformer architectures we implemented the $0^+$-rule for all layers. However, for the Simple3DFormer and the PC Transformer, we had to add custom relevance propagation through the whole model, as the architectures come with several sub-modules such as "local gathering" for the PC Transformer, which are non-trivial to backpropagate through.

**Raw Attention**    We always use the raw attention of the last Transformer block and use bilinear or trilinear interpolation to rescale the attention for image and volume data. For point cloud data, this procedure is more complicated as the PC Transformer projects the embeddings on which the Transformer acts via farthest point sampling and k-nearest neighbor grouping. Thus in each downsampling step, we save which k points are sampled to then use k-nearest neighbor interpolation to cast the attention values for these remaining points back into the input space onto all 1024 original points.

**Rollout Attention**    Same procedure as for Raw Attention but before we interpolate back into the original input space, we use the rollout attention aggregation algorithm over all Transformer modules in the architecture.

**LRP Attention**    As for LRP we use custom relevance backpropagation for the Simple3DFormer and PC Transformer architectures. Based on the relevance scores, we filter the attention of each Transformer module, aggregate the filtered attention with the rollout algorithm, and interpolate the resulting attention back into the input as described for Raw Attention.

### E.2   Adaption of evaluation metrics

In this section, we explain how we adapted the evaluation metrics in our framework to seamlessly work with 3D modalities. All metrics were adapted for point cloud (n,d) and volume (x,y,z) dimensions besides classical image dimensions (w,h,c). We neglected the metrics which did not need any further adaption. All metrics leveraging threshold values expect normalized saliency maps on the observation level. Otherwise, thresholds have to be selected per observation.

**Pixel Flipping**    We compute the Area Under the Curve (AUC) to receive a single score. For point cloud data acts on the single coordinates.

**Region Perturbation**    We compute the AUC to receive a single score. Acts on a 3D kernel for volume data and single points for point cloud data. Compute the AUC to receive a single score.

**Insertion**    Use Gaussian noise for 3D data instead of Gaussian blur for images. Inserting single points for point cloud data and voxels for volume data.

**Deletion**    Deletes single points for point cloud data and voxels for volume data. Compute the AUC to receive a single score.

**Iterative Removal of Features (IROF)**    Compute the Area Over the Curve (AOC) to receive a single score. We use 3D Slic for volume segmentation and KMeans clustering with fixed $k = 16$ clusters for point cloud segmentation. $k = 16$ was determined by visual inspection. See exemplary visualization in Figure 8.

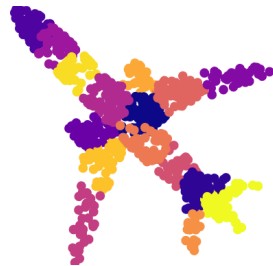

Figure 8: Example of KMeans clustering for point cloud data with k=16.

**Remove and Debias (ROAD)**    We use Gaussian noise for 3D modalities. Compute the AUC to receive a single score.

**Sufficiency**    Use the whole set of saliency maps for similarity comparison and not only the batch the metric is applied to (see Appendix L). For distance calculation between saliency maps, we use squared

Euclidean distance for volume data and standardized Euclidean distance for image and point cloud data due to numerical instability.

**Continuity** We implemented x-axis traversal for volume data along the x-axis with black padding in all dimensions and for the point cloud data by traversing all points along the x-axis position at $(n, d = 0)$ (see Figure 9). As removing points for point cloud data would change the input dimension of the object, we instead map them to the center (0,0,0). We did not observe any OOD behavior by implementing this solution. We use the Pearson Correlation Coefficient (PCC) between traversals to compute a single score.

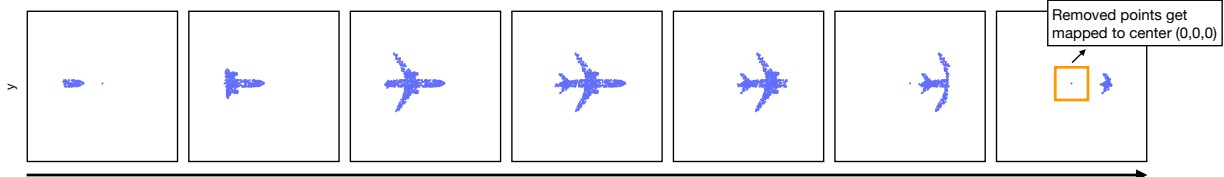

Figure 9: X-axis traversal of point clouds for continuity metric. We can not remove points as this would change the input dimensionality, thus we map them to the center (0,0,0), which is similar to black padding for image and volume data.

**Relative Representation Stability** We use uniform noise ($U(0, 0.05)$) due to numerical stability as Gaussian noise could generate infinity values.

## F Ensuring comparability of results

To ensure fair comparability between XAI methods over all possible experiment set-ups and aggregation levels, we take precautions about the XAI methods, evaluation metrics, and model architectures. Attribution measures the positive or negative contribution of an input feature (e.g. pixel) into the predicted output class of the model. On the contrary, CAM methods only compute positive attribution, and attention highlights all general (or absolute) important input features independent of the output class. However, in practice, attention is only valuable in interpretation if it also highlights features that are used for prediction. New methods such as LA filter the attention to only show such class-relevant attention, and their possible better performance to unfiltered attention can only be shown by evaluating it as positive attribution. Thus we consider only positive attribution for saliency map comparison (also suggested by Zhang et al. (2018)).

Further, we normalize the saliency maps on the observation level as some metrics have nominal thresholds or noise intensities which depend on the scale of saliency maps. As not all metrics compute single scores we have to convert all metrics computing sequences or array of sequences into single scores either via the AUC for Pixel Flipping, Region Perturbation, Selectivity and ROAD, AOC for IROF, or the PCC for SensitivityN and Continuity. All scores are normalized on the metric and dataset level. Score backpropagation-based metrics such as LRP (excluding the $0^+$-rule), DS or DLS, and the CAM methods expect non-negative activation outputs. Thus, we exchanged before the CNN model training all GeLU or leakyReLU activation functions with standard ReLU functions as they output negative values, biasing the XAI method. For the Transformer architectures, however, we keep all activation functions, as well as the skip connections and patchification, as they are central to the architecture. Their potential effect on different attribution methods is part of the benchmark. For CAM methods on the Transformer architectures, we interpolate the reshaped *absolute* cls token, as saliency maps would otherwise often be empty (also recommended by Chefer et al. (2021)).

## G Ranking computation flow chart

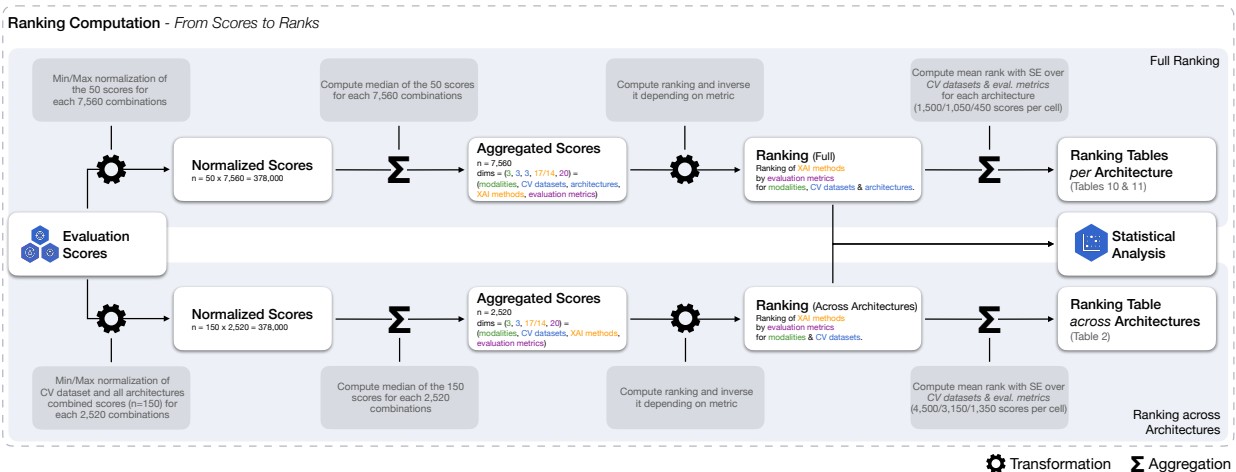

Figure 10: Transformation and aggregation steps from raw evaluation scores to final tables.

Figure 10 shows the transformation and aggregation steps from raw scores to final tables depending if we want to average across architectures or construct tables per architecture. In the calculation of the combinations, it must be taken into account that in the case of the Transformer architectures we have three more XAI methods (attention methods), and in the case of the point cloud modality, we have three fewer XAI methods (excluding CAM methods). In the case of the full ranking, we then have 7,560 combinations of CV datasets, architectures, XAI models, and evaluation metrics based on which we compute 50 scores for each combination, but always the same observations per dataset. If we average across architectures, we have to first normalize the 50 scores per architecture together, as the number of XAI methods differs between CNN and Transformer-based architectures. As we normalize across architectures we end up with 2,520 combinations but 150 scores per combination, which are in total again 378,000 scores. To receive the tables in the final step, we take the mean over the computer vision datasets and evaluation metrics per evaluation criteria, to receive one average rank per XAI method, evaluation criteria, modality, and depending if the ranking is full or across architectures, architecture. Values in the last aggregation step coincide with the number of scores per evaluation criteria, as each of the three criteria contains a different amount of metrics.

## H Metric standard deviation for volume and point cloud data

**a.** Avg. Standard Deviation for Volume Model Architectures and Datasets

| Evaluation Criteria: | Model Architectures | | | Utilized CV Datasets | | |
|---|---|---|---|---|---|---|
| | 3DResNet18 | 3DEffNetb0 | S3DF | AMN | OMN | VMN |
| **Faithfulness** | 3.07 | 3.41 | 3.61 | 3.34 | 3.19 | 3.55 |
| **Robustness** | 3.47 | 3.41 | 3.42 | 3.54 | 3.25 | 3.51 |
| **Complexity** | 0.45 | 0.48 | 0.64 | 0.51 | 0.63 | 0.43 |
| *Weighted Average* | 2.82 | 2.99 | 3.1 | 2.99 | 2.83 | 3.07 |

**b.** Avg. Standard Deviation for PC Model Architectures and Datasets

| Evaluation Criteria: | Model Architectures | | | Utilized CV Datasets | | |
|---|---|---|---|---|---|---|
| | PointNet | DGCNN | PCT | CMA | M40 | SHN |
| **Faithfulness** | 2.9 | 2.97 | 3.55 | 3.23 | 3.07 | 3.1 |
| **Robustness** | 2.52 | 2.74 | 2.91 | 2.9 | 2.8 | 2.48 |
| **Complexity** | 0.72 | 0.51 | 0.29 | 0.42 | 0.59 | 0.51 |
| *Weighted Average* | 2.44 | 2.52 | 2.84 | 2.69 | 2.6 | 2.49 |

Table 9: Average metric standard deviation per model architectures and utilized datasets for **a.** volume and **b.** point cloud modalities.

# I Ranking-bias through metric subsets

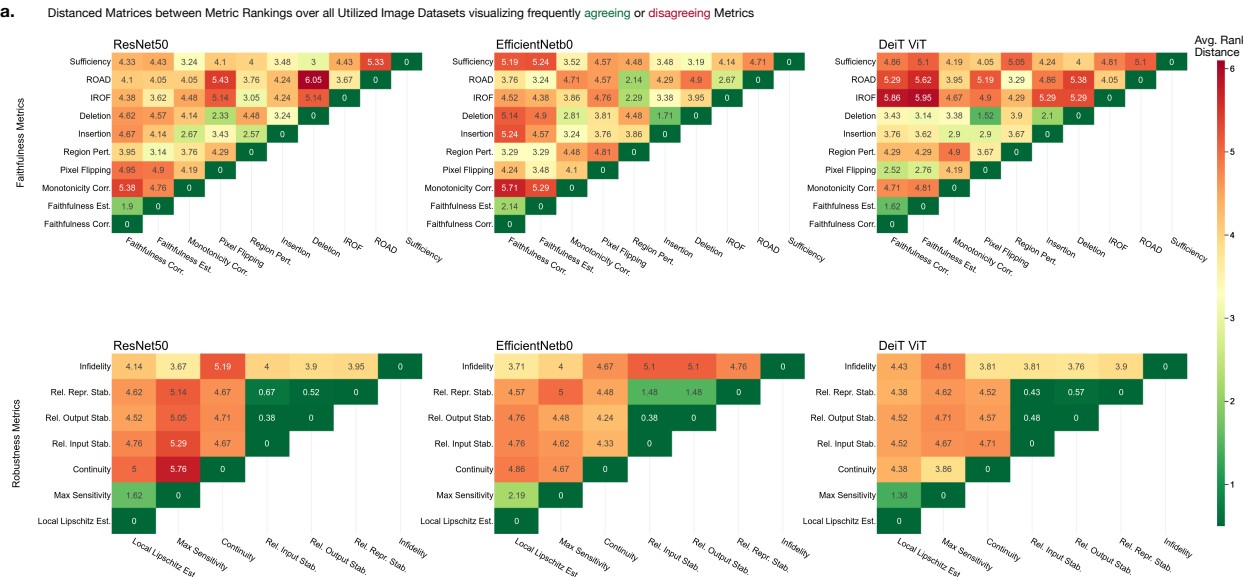

Figure 11: Average Euclidean ranking distance between metric pairs for model architectures and the faithfulness and robustness criteria. More often agreeing metric pairs in their rankings appear more green, and disagreeing pairs more red.

In scenarios of high disagreement among metrics, potential biases could arise when considering only a limited subset of metrics, a concern we have raised in relation to other studies. Such scenarios further underscore our large-scale experimental design as it prevents undetected biases that could result from the selective use of individual metrics, intentional or accidental. Figure 11 shows the Euclidean distance, averaged over all attribution methods and datasets, between the ranking of two metrics for all model architectures and the faithfulness and robustness criteria. We observe that the general mean distance or disagreement between two metrics is around 4 ranks. We further observe around 8-10 outlier pairs for faithfulness and 2-5 for robustness, which have either substantially higher agreement ($\sim 2$ ranks) or higher disagreement ($\sim 5.5$ ranks). By first selecting a favorable metric and then pairing it with strongly similar ranking metrics based on this distance matrix (e.g. Faithfulness Estimate with Faithfulness Correlation), it is in theory possible to selectively pair similar ranking metrics to deliberately skew results towards a favorable outcome. However, this is only possible for a very small subset of metrics. Also the likelihood of any such "extremist subgroups" unduly influencing our large-scale study is small. The closest example of such subgroups would be the Relative Stability metrics for robustness.

## J   Disagreement between metrics for XAI methods with high SD

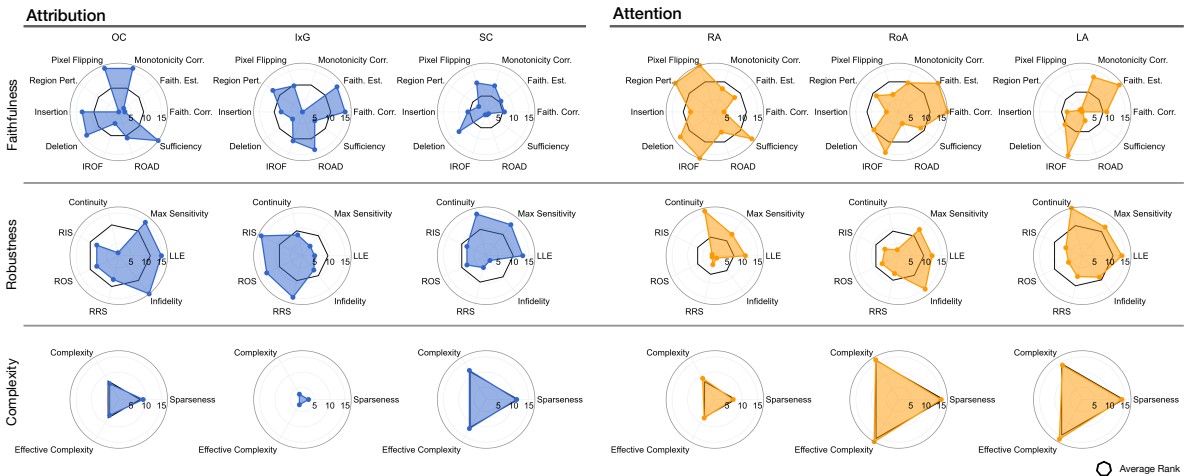

Figure 12: Rankings of all metrics for the three attention and attribution methods with the largest proportion of variance larger than random ranking variance (see Equation 3. Specific for the ImageNet dataset and ResNet50 model architecture.

## K   Additional ranking tables

### K.1   Full ranking table with standard errors

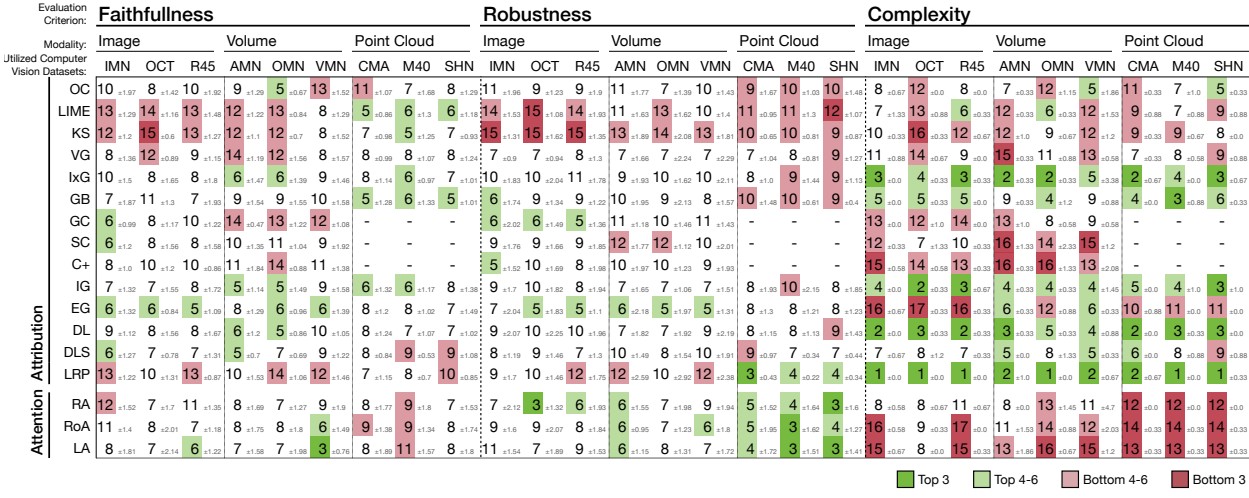

Table 10: Full ranking table for all XAI methods and CV datasets with standard error (SE).

## K.2 Ranking table CNNs only

| Evaluation Criteria: | Faithfullness | | | Robustness | | | Complexity | | |
|---|---|---|---|---|---|---|---|---|---|
| Modality: | Image | Volume | Point Cloud | Image | Volume | Point Cloud | Image | Volume | Point Cloud |
| OC | 8.5 | 5.5 | 9 | 11.5 | 5.5 | 4.5 | 9 | 10.5 | 7.5 |
| LIME | 14 | 9.5 | 2.5 | 14 | 12.5 | 11 | 6 | 8.5 | 7.5 |
| KS | 13 | 9.5 | 7 | 13 | 14 | 9 | 13.5 | 8.5 | 10 |
| VG | 11.5 | 11 | 9 | 6 | 2 | 2 | 10 | 13 | 9 |
| IxG | 10 | 2.5 | 5 | 11.5 | 9.5 | 7 | 3.5 | 1.5 | 3 |
| GB | 8.5 | 7 | 1 | 10 | 8 | 8 | 2 | 5 | 5.5 |
| GC | 3 | 14 | - | 2 | 11 | - | 12 | 7 | - |
| SC | 5 | 8 | - | 8 | 9.5 | - | 7.5 | 14 | - |
| C+ | 4 | 13 | - | 4.5 | 7 | - | 11 | 12 | - |
| IG | 6 | 1 | 5 | 8 | 3 | 4.5 | 5 | 4 | 4 |
| EG | 1 | 5.5 | 11 | 2 | 1 | 10 | 13.5 | 10.5 | 11 |
| DL | 7 | 2.5 | 2.5 | 8 | 4 | 4.5 | 1 | 3 | 2 |
| DLS | 2 | 4 | 9 | 4.5 | 5.5 | 1 | 7.5 | 6 | 5.5 |
| LRP | 11.5 | 12 | 5 | 2 | 12.5 | 4.5 | 3.5 | 1.5 | 1 |

Per modality: ▇ Top 1  ▇ Top 2-4  ▇ Bottom 2-4  ▇ Bottom 1

Table 11: Ranking of the average rank over CNN architectures, datasets, and all evaluation metrics of the respective criteria, for each XAI method and modality (i.e. the rank of OC on image is based on $3*2*10 = 60$ ranks). Coloring coincides with top and bottom positions as no attention methods can be applied to CNN architectures.

## K.3 Ranking table Transformer only

| Evaluation Criteria: | Faithfullness | | | Robustness | | | Complexity | | |
|---|---|---|---|---|---|---|---|---|---|
| Modality: | Image | Volume | Point Cloud | Image | Volume | Point Cloud | Image | Volume | Point Cloud |
| OC | 13 | 9 | 11.5 | 12.5 | 12 | 13 | 11.5 | 8.5 | 6.5 |
| LIME | 16.5 | 15 | 2 | 15.5 | 16 | 14 | 9 | 8.5 | 5 |
| KS | 15 | 14 | 4.5 | 17 | 17 | 10 | 17 | 12.5 | 9 |
| VG | 11.5 | 11 | 8.5 | 1.5 | 2 | 7 | 13.5 | 10 | 6.5 |
| IxG | 9.5 | 3 | 1 | 7.5 | 10 | 10 | 1 | 1 | 3 |
| GB | 5 | 5 | 3 | 10 | 13 | 12 | 4.5 | 5 | 4 |
| GC | 9.5 | 16.5 | - | 12.5 | 10 | - | 10 | 14 | - |
| SC | 6.5 | 11 | - | 6 | 15 | - | 6 | 16.5 | - |
| C+ | 14 | 16.5 | - | 15.5 | 10 | - | 13.5 | 16.5 | - |
| IG | 6.5 | 2 | 6 | 5 | 5.5 | 8 | 3 | 3 | 2 |
| EG | 1 | 1 | 7 | 1.5 | 1 | 6 | 15.5 | 7 | 12 |
| DL | 11.5 | 4 | 4.5 | 10 | 7.5 | 10 | 4.5 | 2 | 1 |
| DLS | 3 | 6.5 | 8.5 | 3 | 7.5 | 5 | 7 | 6 | 8 |
| LRP | 16.5 | 13 | 13.5 | 14 | 14 | 4 | 2 | 4 | 10.5 |
| | | | | | | | | | |
| RA | 8 | 11 | 10 | 4 | 3.5 | 3 | 8 | 12.5 | 10.5 |
| RoA | 4 | 6.5 | 11.5 | 10 | 3.5 | 2 | 15.5 | 11 | 14 |
| LA | 2 | 8 | 13.5 | 7.5 | 5.5 | 1 | 11.5 | 15 | 13 |

Per modality: ▇ Top 1  ▇ Top 2-4  ▇ Bottom 2-4  ▇ Bottom 1

Table 12: Ranking of the average rank over Transformer architectures, datasets, and all evaluation metrics of the respective criteria, for each XAI method and modality (i.e. the rank of OC on image is based on $3*1*10 = 30$ ranks). Coloring coincides with top and bottom positions.

## L Shortcomings of evaluation metrics in practice

While all metrics are theoretically very well founded, we observed for some metrics shortcomings in applications:

Casting saliency maps from 64-bit to 32 or 16-bit to save disk space in such large evaluations is not recommended, as our experiments showed that even 32-bit precision can lead to numerical instability, resulting in all-zero saliency maps and *nan* or *inf* evaluation scores.

Sufficiency evaluates the likelihood that observations with the same saliency maps also share the same prediction label. In practice, this requires several saliency maps from observation with the same prediction label. While this works well on datasets with a small number of labels and balanced sampling, for datasets like IMN with 1000 labels, the probability is almost zero that at least 5-10 sampled observations in a set of sizes 50 or 100 have the same label (see Appendix O).

Sequence outputting metrics that alter the input space, such as Pixel Flipping, Region Perturbation, or ROAD, are only limited suitable for binary prediction tasks. When the input object is too noisy/perturbed to predict accurately, the probability for each class is 0.5 resulting in sequences converging against 0.5 and not 0. While the resulting AUC (or AOC in the case of Region Perturbation) can be compared between XAI methods within this task, between tasks the AUC would be biased as the area for the binary task would always be larger.

ROAD scores are arrays of binary sequences which are averaged to one sequence. The amount of noise has to be carefully tuned (also depending on the underlying model) as otherwise, all binary sequences in the array are only 0 or 1.

Local Lipschitz Estimate approximates the Lipschitz smoothness through several forward passes of a batch of observations. In application, this results in a large amount of RAM used (depending on modality) if the approximation should be stable. While the computation is relatively fast on a GPU, stable approximations exceed 40GB of VRAM by far and have to be partitioned. For the Transformer architectures, computation on the CPU for our amount of data was too slow to be feasible.

Effective complexity uses a nominal threshold value to determine attributed features. Even through normalization of the saliency maps, the threshold value can have a large effect on the results, differing between observations, and we would suggest tuning it per dataset.

IROF superpixel segmentation can result in very defined or binary structures such as in the AMN dataset in only two superpixels (object and background), ignoring finer structures.

As elaborated, all complexity metrics flatten the input object treating it as a vector and ignoring spatial dependencies.

## M    Ranking distance between model architectures

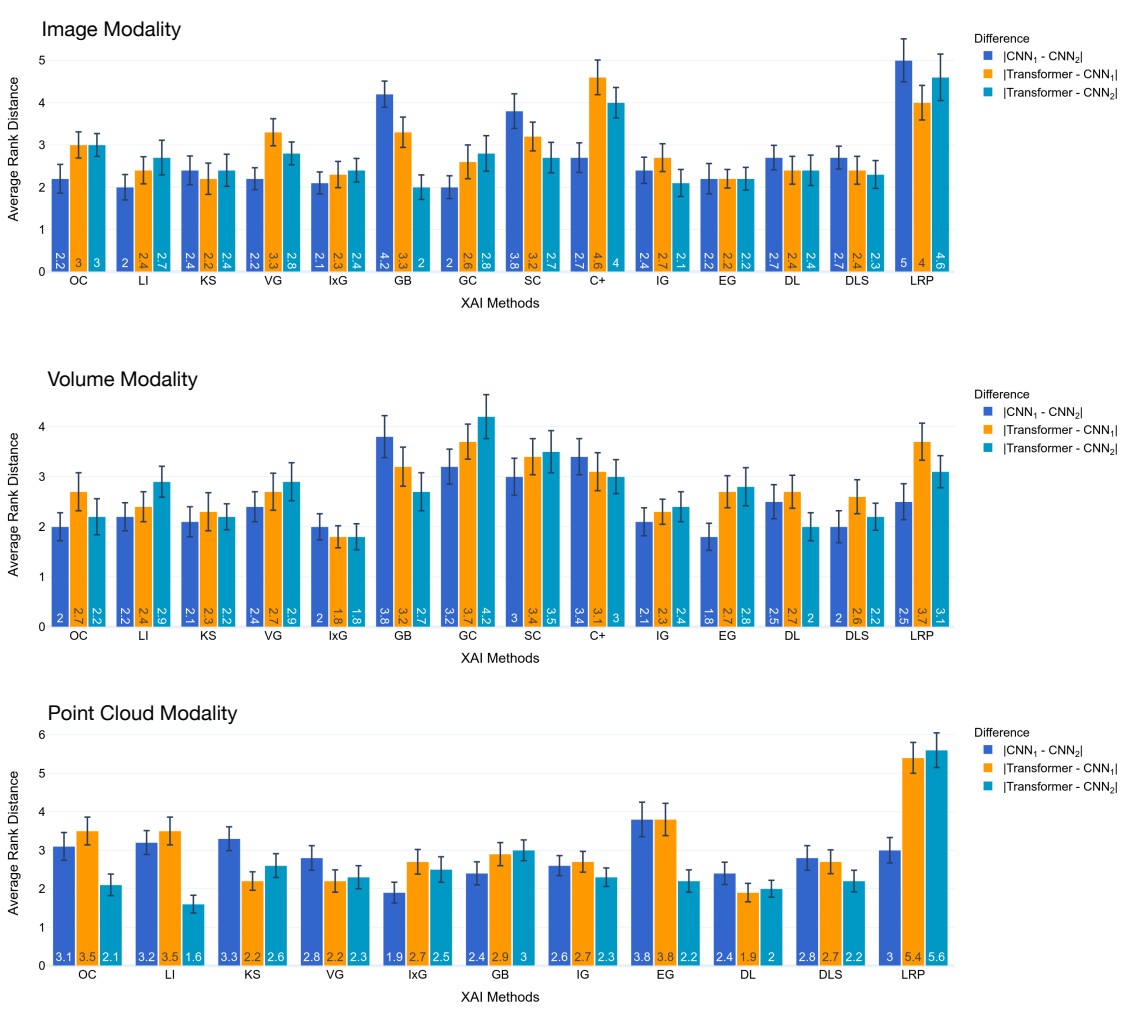

Figure 13: Average distance between ranks of XAI methods on different model architectures for all modalities.

# N Differences in ranking order between model architectures

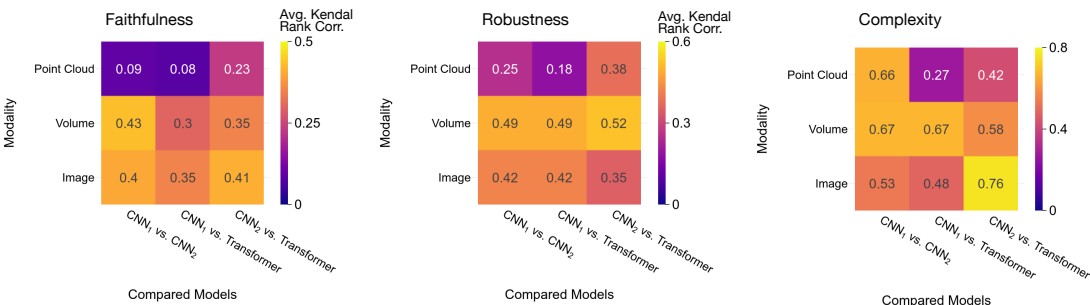

**a.** Average Rank Correlation of Evaluation Metrics between Models

Figure 14: Kendall's-$\tau$ rank correlation between model architectures averaged over datasets and faithfulness criteria.

We compare the difference in faithfulness rankings of attribution methods between CNN and Transformer architectures, as biased methods should be less faithful to the model. To this end, we compute the Kendals-$\tau$ rank correlation between each of the three architectures per dataset and compute their average correlation per modality (see Figure 14). We observe a positive correlation between all rankings. For the point cloud modality, however, the correlation is significantly lower than for the other two modalities, indicating less similar rankings between model architectures. For volume and image modality, the similarity between CNN architectures is generally higher.

# O  Score distributions of evaluation metrics for all datasets

## O.1  Image modality

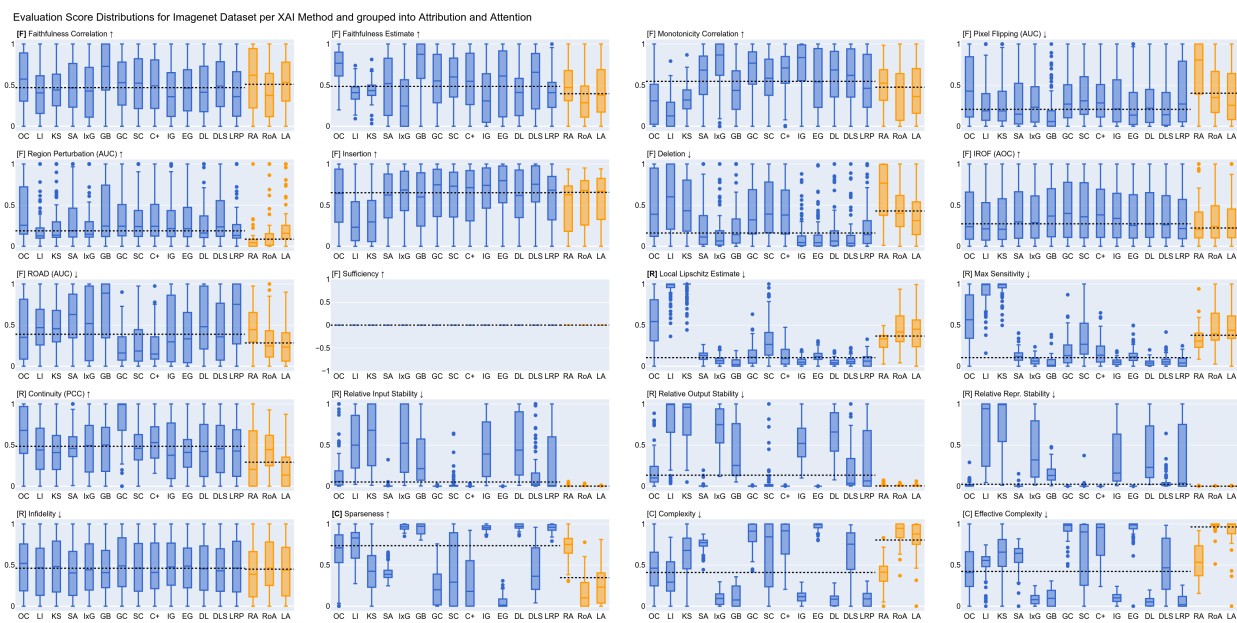

Figure 15: Score distributions of all evaluation metrics for each XAI method. Scores are normalized also for the Continuity PCC, as a negative correlation is worse than no correlation.

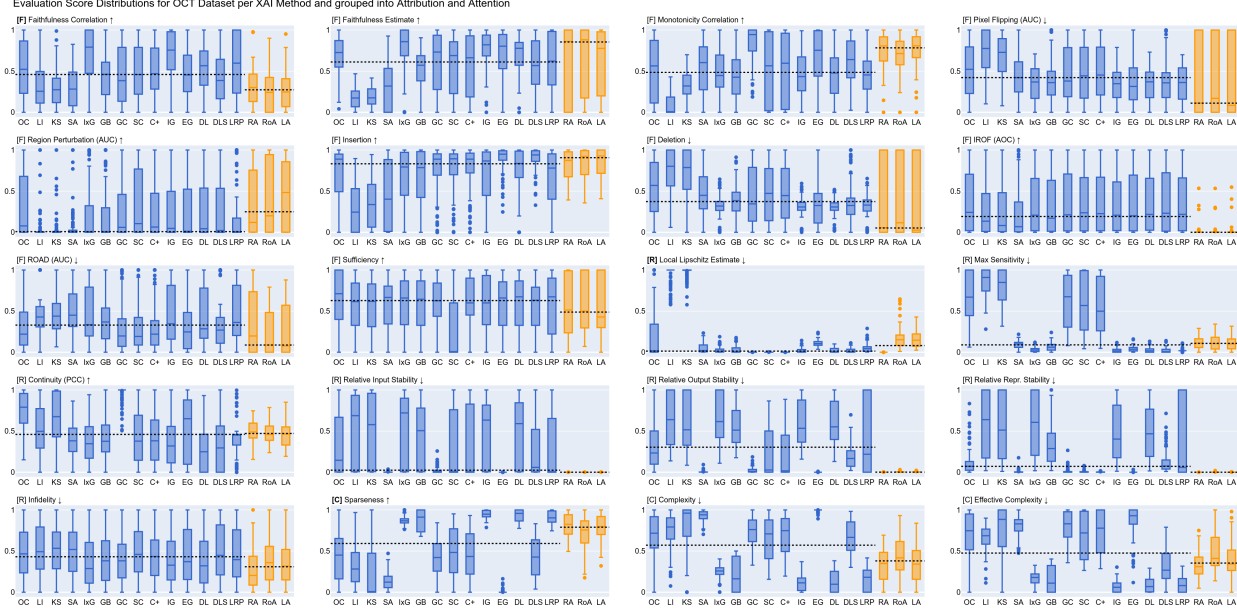

Figure 16: Score distributions of all evaluation metrics for each XAI method. Scores are normalized also for the Continuity PCC, as a negative correlation is worse than no correlation.

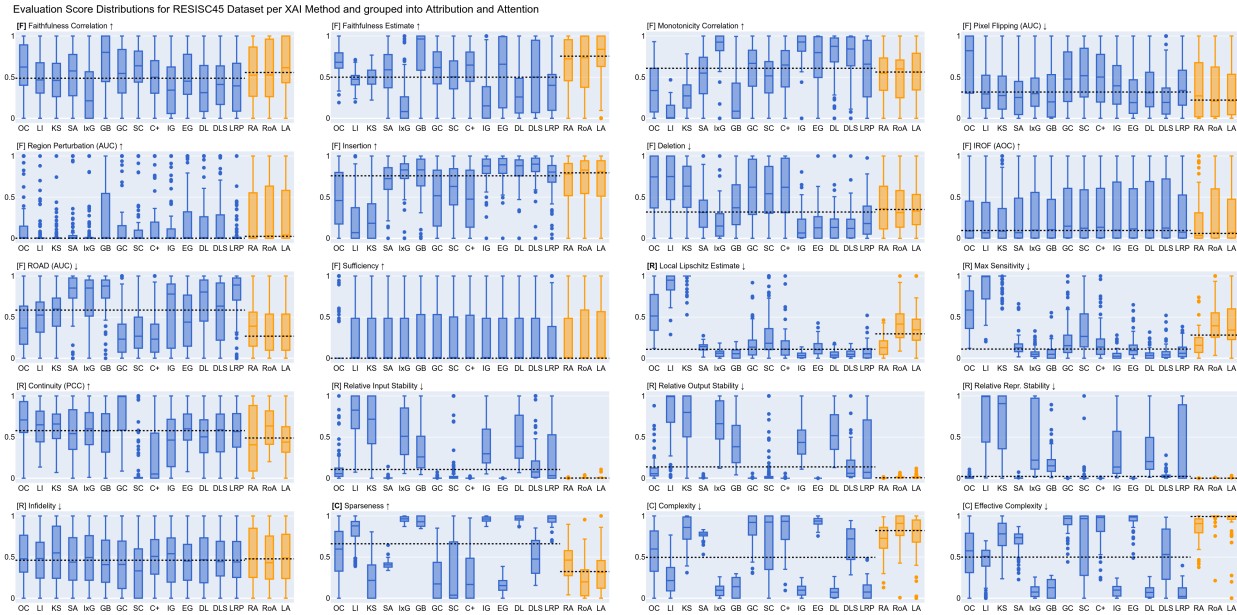

Figure 17: Score distributions of all evaluation metrics for each XAI method. Scores are normalized also for the Continuity PCC, as a negative correlation is worse than no correlation.

## O.2 Volume modality

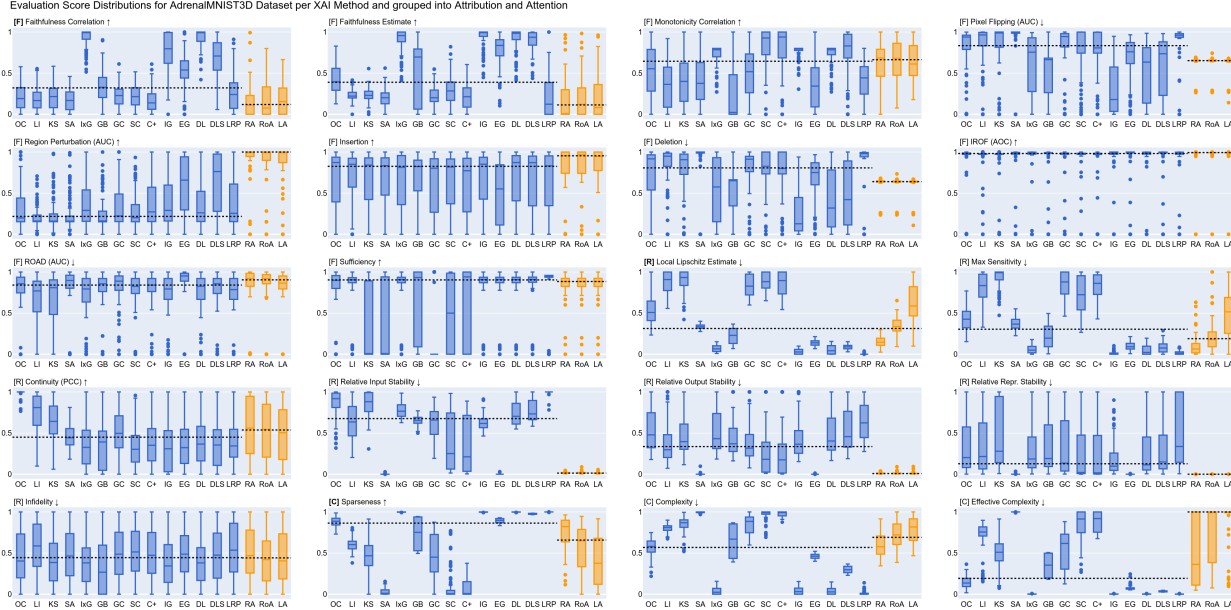

Figure 18: Score distributions of all evaluation metrics for each XAI method. Scores are normalized also for the Continuity PCC, as a negative correlation is worse than no correlation.

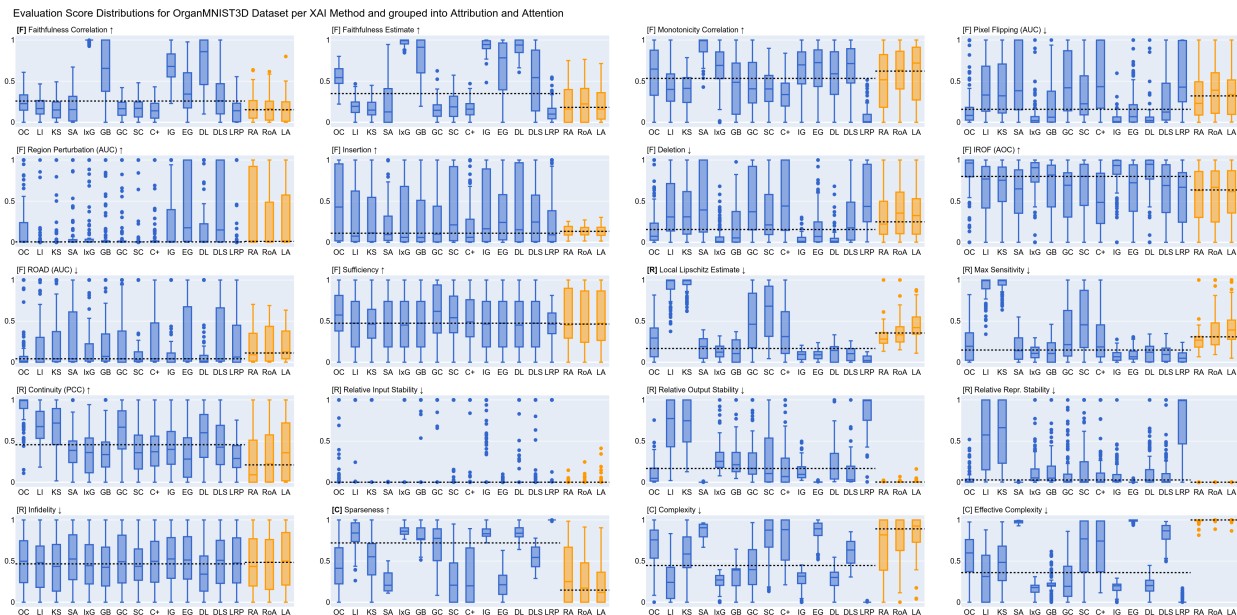

Figure 19: Score distributions of all evaluation metrics for each XAI method. Scores are normalized also for the Continuity PCC, as a negative correlation is worse than no correlation.

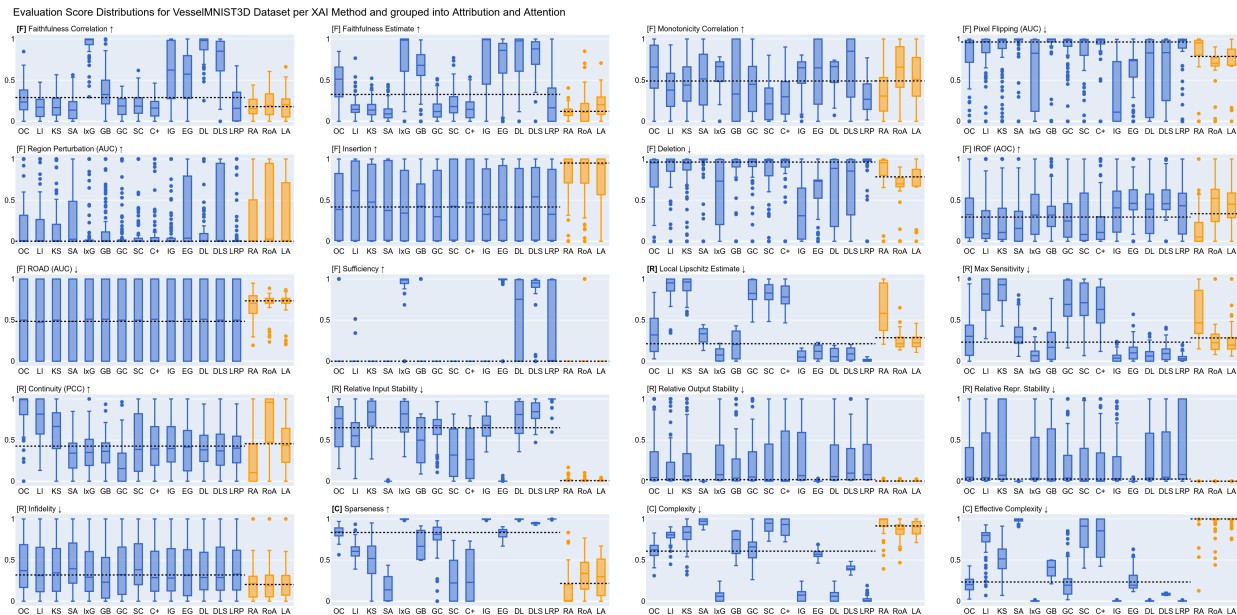

Figure 20: Score distributions of all evaluation metrics for each XAI method. Scores are normalized also for the Continuity PCC, as a negative correlation is worse than no correlation.

## O.3 Point cloud modality

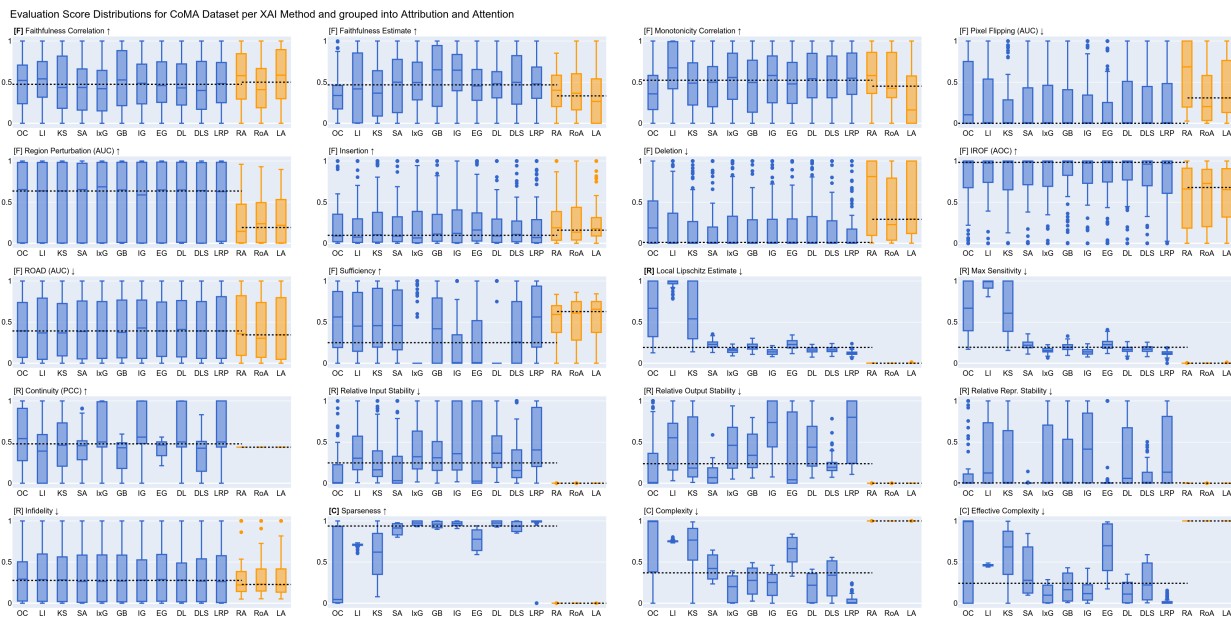

Figure 21: Score distributions of all evaluation metrics for each XAI method. Scores are normalized also for the Continuity PCC, as a negative correlation is worse than no correlation.

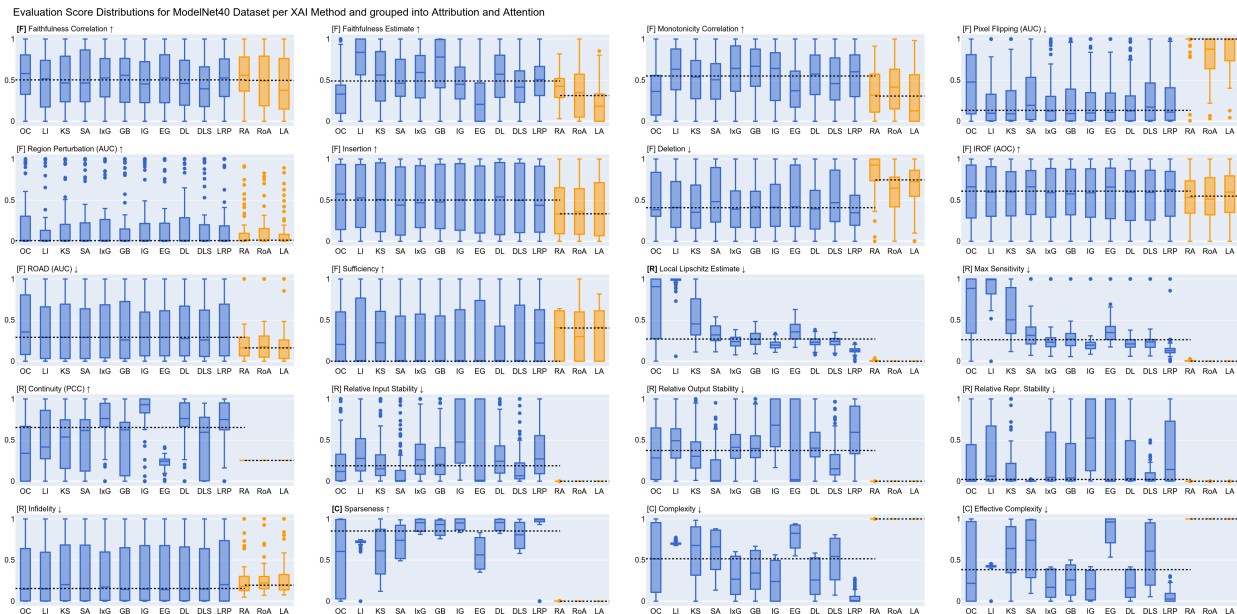

Figure 22: Score distributions of all evaluation metrics for each XAI method. Scores are normalized also for the Continuity PCC, as a negative correlation is worse than no correlation.

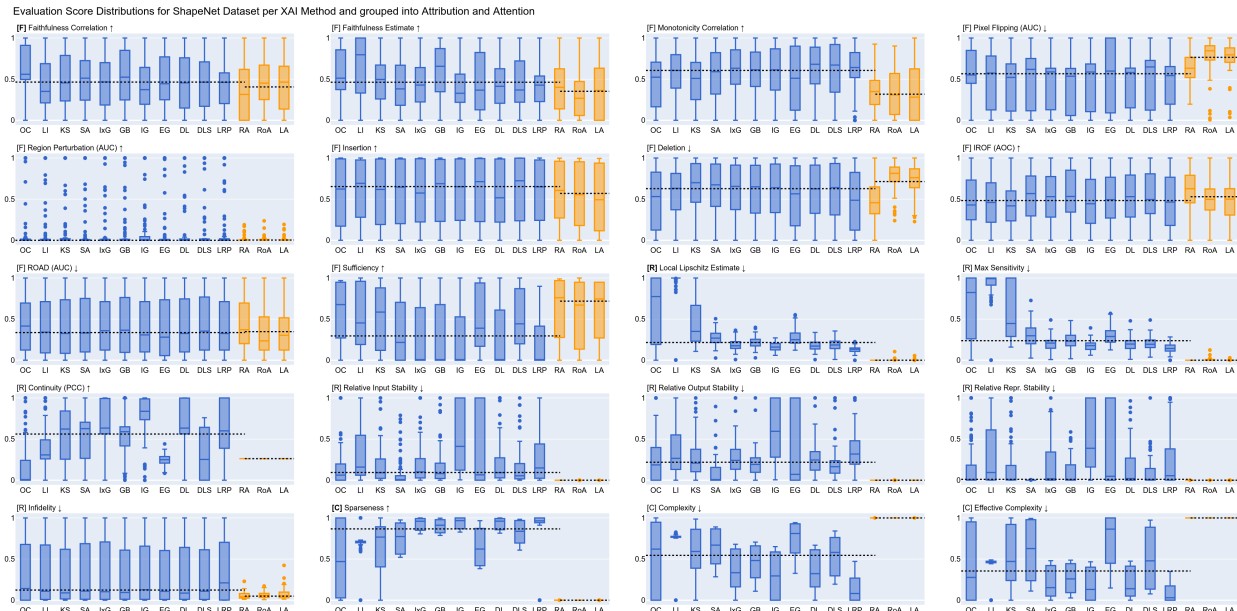

Figure 23: Score distributions of all evaluation metrics for each XAI method. Scores are normalized also for the Continuity PCC, as a negative correlation is worse than no correlation.

