# OpenReview forum: "Navigating the Maze of Explainable AI: A Systematic Approach to Evaluating Methods and Metrics"
_TMLR — Withdrawn by Authors_

### Review · Reviewer_krZA · 2024-04-17

**Summary Of Contributions:**

This paper introduces a benchmark, namely LATEC, that evaluates 17 prominent XAI methods using 20 distinct metrics. The authors propose a robust evaluation scheme based on LATEC, so as to obtain a relatively reliable evaluation rankings. Furthermore, the authors evaluate various XAI methods and obtain various new insights for XAI evaluation.

**Audience:**

Yes

**Broader Impact Concerns:**

The authors should provide a Broader Impact Statement.

**Claims And Evidence:**

No

**Requested Changes:**

Critical Changes

1. A comprehensive reevaluation of the benchmarking framework used for assessing XAI methods. Specifically, the approach to integrating various metrics to form a 'robust' evaluation needs to be rethought to avoid contradictions. Currently, the methodology lacks a clear rationale for the selection and averaging of metrics, which could lead to unreliable assessments. This is fundamental to the credibility of the research findings.
2. The paper should include explicit mathematical formulas for all computational procedures and results presented. Mathematical modeling is essential for clarity and precision in scholarly work, especially in evaluative studies. Providing these formulas will help readers understand the research methodology without ambiguity and ensure the reproducibility of the results.

Strengthening Changes

1. Additionally, there is a significant need to enhance the quality of the writing in the manuscript, as discussed in weaknesses. It is essential to elevate the paper to a publishable standard within such a prestigious journal.

**Strengths And Weaknesses:**

[Strengths]

1. This paper provides a relatively comprehensive evaluation of the performance of various XAI methods, taking into account different types of data and model architectures.

2. This paper obtains various new findings for XAI evaluation, some of which have the potential to offer new perspectives for the development of XAI methods and are likely to stimulate further discussions and advancements in the field.

[Weaknesses]

Although this paper presents some new conclusions for XAI evaluation, the lack of clarity in its presentation raises questions about the credibility of its conclusions. Here are some specific issues and recommendations:

1. Using the average rating of different metrics as a so-called "robust" evaluation is problematic. Metrics are often designed for different data types, and their underlying implications can even be contradictory. Merely averaging different metrics to achieve a "robust" rating completely ignores these issues. In extreme cases, if I arbitrarily design 100 metrics, could they also be incorporated into your evaluation framework?

2. Lack of Mathematical Modeling. For an evaluative paper, the absence of any formulas is unreasonable. While the authors describe evaluation formulas verbally (e.g., "we computed the standard deviation between metric rankings mean aggregated for each underlying design parameter," "Proportion of accepted one-sided Levene-Tests for significant (α = 0.1) smaller ranking variance compared to the variance of an entire random ranking"), there is still a need for mathematical modeling to allow readers to fully understand the intentions, rather than guessing from textual descriptions. Therefore, I suggest that the authors should provide explicit computational formulas for all presented results (including tables and figures) and clearly define each variable in these formulas.

3. Due to the lack of clear formulation of evaluation formulas, I do not understand why the evaluation of complexity in Figure 1a is so uniform. Are the results from different complexity metrics for the same XAI method identical?

4. The authors hastily draw conclusions from the results without clear explanations of why certain experimental outcomes support specific conclusions. Therefore, I suggest that the authors provide more discussion and clearer analysis for each experimental result to substantiate their conclusions. For example, in the analysis of Figure 1a, how is ranking-agreement or ranking-disagreement defined and distinguished? What exactly does “one strong outlier for DLS” refer to?

5. The paper frequently employs statistical methods but seems to assume that all readers are very familiar with statistical terms, hence it neither provides the calculation formulations for these metrics nor explains the meanings of important parameters. This greatly affects the readability of the paper. For instance, the authors use one-sided Levene’s Test to check rank-variance, but what does α signify? Is it a threshold? Please clarify its physical meaning as not all readers may have a strong background in statistics.

6. There are several errors in the text. For example, there are no results for the OC method in Figure 2a, yet the authors claim "as the 'Average' faithfulness for OC and IG shows in Figure 2 (a.)."

7. Lastly, the authors should conduct a more thorough review of the manuscript to eliminate grammatical and spelling errors. For example, in the discussion of prior work, there is inconsistent verb tense usage, such as the past tense ("showed") and the present tense ("can") used in the same context: "as several other works ... showed that advancing methods ... can, in general, improve results.” Such inconsistencies can distract readers and detract from the credibility of the paper.

In sum, I believe that the manuscript currently does not meet the publication standards of TMLR.

---

> ### Author Response · Authors · 2024-05-03
> **Answer to reviewer krZA**
>
> We thank reviewer krZa for the extensive and elaborate feedback. While we address the concern regarding **W1** in our response to all reviewers, all other raised questions are addressed point by point below.
>
> **Answer to W2:** Thank you for pointing out the lack of mathematical formality. We agree that formulas for the mathematical modeling behind the provided results, and a direct link to each figure and table, would greatly improve their comprehensibility. Thus we introduced mathematical notation and formulas throughout the whole manuscript to substantiate our approach. Each formula is highlighted with a reference to the respective figure which shows the output of the respective computation. Similarly, each figure specifically refers to the respective formula for making calculations as transparent and comprehensible as possible.
>
> **Answer to W3:** Yes, for the displayed set of design parameters, the complexity metrics all rank the different XAI methods, LIME, IG, GC, and DLS, equally. There is no formula at use here except the selection of the ranking in this specific setting (i.e. Image modality = ImageNet dataset, and model architecture = ResNet50 ). As we describe in the paper, this observation is also one of our motivations to look deeper into the complexity metrics. In the manuscript, we now describe the displayed design parameters behind each of the line plots. Additionally, the inclusion of mathematical notation clarifies the interpretation of data presented in the plots.
>
> **Answer to W4:** Thank you for your feedback on the clarity of our manuscript. Regarding the disagreement and agreement of the rankings between various metrics: In our study, we assess how different metrics, approximating the same criteria, rank XAI methods under identical design parameters (i.e. modality, dataset, and model architecture). In our metric analysis, we compare how each metric belonging to one criterion ranks a specific XAI method in comparison to all other 16 XAI methods. By comparing the computed ranks between all metrics for this one XAI method, we can determine agreement (i.e. consensus) or disagreement among metrics regarding how to rank this XAI method. For example in Figure 2 a., metrics largely agree on the ranking of LIME under the faithfulness criterion (around rank 12), whereas they show significant disagreement for GC. The quantification of disagreement beyond qualitative analysis is discussed in the answer to all reviewers above.
>
> Regarding the previous vague description of "one strong outlier for DLS" we have now clarified that while most metrics rank DLS highly, the IROF metric ranks it substantially lower, highlighting a specific discrepancy.
>
> Throughout the whole manuscript, we have enlarged our descriptions and explanations to enhance clarity and improve comprehensibility.
>
> **Answer to W5:** Thank you for pointing out that statistical notation is probably not familiar to everyone, making the paper less inclusive. The term "alpha" represents the significance level at which a test result is accepted, a convention widely used across disciplines that employ statistical data analysis. Indeed, this significance level acts as a threshold: if the computed p-value — the probability that an observed effect occurs by chance — is below this threshold, we reject our null or baseline hypothesis (i.e. in the case of the Levene Test that the variance between ranks from different metrics is equal or larger than the variance of a random ranking), suggesting that the effect is statistically significant.
>
> Based on the feedback, we now provide detailed descriptions of all parameters used in the test and added rationales behind the test procedure. Furthermore, the inclusion of mathematical notation and formulas throughout the manuscript enhances clarity regarding the application of the test in our analysis.
>
> **Answer to W6:** Indeed this is a mistake and it should be GC, not OC. Thank you for pointing this out. Besides this mistake, could you please point out the other “several errors” in the text you noticed?
>
> **Answer to W7:** Thank you for pointing this out, we thoroughly reviewed the manuscript for such errors, and corrected the tense of the paper so that the full paper is now written consistently in the present tense.

---

### Review · Reviewer_pcne · 2024-04-21

**Summary Of Contributions:**

This paper introduces LATEC, a large-scale benchmark for evaluating eXplainable Artificial Intelligence (XAI) methods. LATEC assesses 17 popular XAI methods across various input modalities and network architectures, employing diverse metrics for comprehensive evaluation. Utilizing mean aggregation, the paper consolidates these evaluation results to derive the average rankings of the 17 XAI methods. Besides, this paper offers some findings and recommendations for the selection and utilization of XAI methods.

**Audience:**

Yes

**Broader Impact Concerns:**

Not applicable.

**Claims And Evidence:**

No

**Requested Changes:**

**Critical for Acceptance:**

1.It is imperative to comprehensively exhibit the technical validity of employing mean aggregation to amalgamate evaluation outcomes across varied metrics.

2.A thorough discussion on the notable concern regarding the standard deviation of rankings is warranted.

3.A more methodical categorization of evaluation metrics, accompanied by cogent rationale, is indispensable.

**Strengths And Weaknesses:**

**Strengths**

1.The research problem of this paper (i.e., a reliable evaluation for XAI methods) is a crucial yet open challenge within the XAI field. It helps practitioners in selecting appropriate XAI methods.

2.Compared to previous work, this paper conducts a more extensive evaluation on popular XAI methods, which covers diverse metrics, various input modalities and network architectures. It is valuable to do such a large-scale investigation.

3.This paper offers some findings and recommendations, some of which may be helpful for XAI researchers. For example, rankings of XAI methods typically generalize well over datasets but can highly depend on input modalities.

**Weaknesses**

1.My main concern is the technical correctness of simply using mean aggregation to incorporate diverse metrics. From my opinion, the methodology is equivocal and lack technical soundness. For example, consider the metrics Monotonicity Correlation and Insertion, both falling under the faithfulness category, but actually evaluate XAI methods from a significantly different perspective. I seriously doubt whether it is reasonable to simply average the evaluation results(rankings) on such disparate metrics.

While the authors mention a weighted aggregation approach to enhance the methodology, I remain skeptical about its efficacy. Reliably determining subgroup categorizations and their respective weights presents a formidable challenge.

2.When employing mean aggregation to derive average rankings, a significant concern arises regarding the standard deviation of rankings associated with each XAI method. For instance, if a particular XAI method exhibits a high average ranking but also a substantial standard deviation, indicating it's "good but unstable," how should one gauge its overall performance?

Additionally, determining the significance of a standard deviation poses another question: what criteria to define the magnitude of a standard deviation? In essence, how large is considered "large" when referring to a standard deviation?

3.Regarding the categorization of evaluation metrics:
(1) The criteria for categorization are not clearly described. Could you provide definitions for the "Faithfulness," "Robustness," and "Complexity" categories?
(2) I have reservations about the current categorization. For instance, I believe the "Infidelity" metric would be better suited under the "Faithfulness" category rather than "Robustness."

4.From my perspective, it would be more insightful to conduct direct comparisons among XAI methods rather than relying solely on average rankings. For instance, investigating the distinctions between XAI explanations generated for Transformer and CNN architectures is more pertinent than comparing average rankings across these two architectures.

---

> ### Author Response · Authors · 2024-05-03
> **Answer to reviewer pcne**
>
> Dear reviewer pcne, thank you for the extensive and insightful feedback. We answer the raised questions **W1** and **W2** in the answer to all reviewers above. They also resulted in significant changes to the manuscript. All other questions are addressed below.
>
> **Answer to W3:** Thank you for pointing out the uncertainty in definition and allocation regarding the evaluation criteria in the manuscript. In response, we added more detailed definitions of the three criteria as well as a description of the mechanisms metrics used to approximate them, with references to other publications (see Section 2: Evaluation metrics).
>
> While almost all metrics can be strictly categorized to one criterion we use Infidelity with a noisy baseline which evaluates faithfulness and robustness simultaneously. Explanations that minimize such Infidelity under varying perturbations in the input space are called “novel explanations” by the authors of the original paper. Our early experiments showed that with context-dependent added noise, the metric is more in line with the other robustness metrics than the faithfulness metrics. Thus we categorized it as a robustness metric. However, the LATEC framework provides the flexibility to remove the noise component and categorize the metric under the faithfulness category if desired. We have now emphasized the significance of the varying perturbation aspect of Infidelity in our experiments, with a detailed description in Appendix Section D.
>
> **Answer to W4:** We agree that direct comparisons between specific architectures like Transformers and CNNs can yield valuable insights. We would like to highlight that we already include the direct comparisons in the manuscript, e.g. between datasets (Appendix K.1) or between model architectures (Appendix K.2 for CNNs and K.3 for Transformer architectures). We agree that both perspectives—direct comparisons and average rankings—are essential for a comprehensive analysis. In the main manuscript, we show the most insightful results, which include both specific analyses for individual parameter behaviors (e.g. modality, datasets, or model architectures) and general trends across e.g. different model architectures, ensuring a thorough examination that addresses both specificity and generalizability. This structure allows us to focus on the most valuable insights without ignoring the large amount of more specific results, which can be looked up if the reader is interested.
>
> We recognize that the placement of pointers to the detailed tables within the manuscript was not ideal. To address this, we have now incorporated direct references to the results in the main figures and tables for improved clarity and accessibility.

---

### Review · Reviewer_U621 · 2024-04-24

**Summary Of Contributions:**

This manuscript offers a detailed evaluation of machine learning explanation methods, highlighting their performance in terms of faithfulness, robustness, and generalizability across different data modalities. It showcases a comprehensive benchmarking exercise using the LATEC dataset to compare XAI methods and provide practical insights for practitioners. However, the work criticizes the current evaluation metrics for their inconsistencies and subjectivity, proposing the need for new metrics that better encompass both quantitative and qualitative aspects. The study also acknowledges limitations, such as excluding non-traditional XAI methods and certain data types, which may affect the generalizability of its findings.

Specifically, this work pinpoints inconsistencies and the risks these pose to both practitioners and researchers, suggesting that the evaluation metrics themselves might need rethinking. It's noteworthy that there is a call for the development of new metrics that bridge the quantitative-qualitative evaluation gap, and a considered reflection on the subjectivity embedded in human conception of complexity. However, the work reveals some self-imposed limitations in the scope of their benchmarks—ignoring many XAI approaches that are less representative, video data, graph data, textual data, and certain critical evaluation criteria which could have provided a richer, more complete analysis. Such limitation potentially restricts the generalizability of the findings to broader real-world applications.

**Audience:**

Yes

**Claims And Evidence:**

Yes

**Requested Changes:**

Please try to address the issues above.

**Strengths And Weaknesses:**

Pros:
- This work offers practical takeaways for the application of XAI, helping practitioners choose the most effective methods based on robust, detailed comparisons and results derived from the LATEC framework/benchmark.
- This study reveals important discrepancies and conflicts in the metrics used for evaluating XAI methods, which is crucial for advancing the understanding and development of more accurate evaluation tools in this field.

Cons:
- The benchmarks restrict their focus to certain data modalities which are considered to have unique, non-overlapping characteristics, excluding others like video/graph/textual data which limits the broad applicability of the results.
- More unconventional post-hoc XAI methods like symbolic representations or metamodels were not included, which may limit insights into potentially valuable approaches.
- Important evaluation criteria such as localization and axiomatic properties weren't included because they either require specific conditions like ground truth bounding boxes or aren’t applicable to all XAI methods.
- Some of recent studies in the relevant lines of research haven't been discussed in this work. For faithfulness evaluation of XAI methods, I suggest discussing this work: M4: A Unified XAI Benchmark for Faithfulness Evaluation of Feature Attribution Methods across Metrics, Modalities and Models, NeurIPS Datasets and Benchmarks track, 2023

---

> ### Author Response · Authors · 2024-05-03
> **Answer to reviewer U621**
>
> We appreciate the insightful feedback from reviewer U621 and have addressed all raised questions point-by-point below.
>
> **Answer to Cons 1:** Thank you for your feedback on the data modalities included in our study. At this point, the scope of our paper is already unprecedentedly larger than any other work. The only work focusing on more than one modality by Li et al. [NeurIPS, 2023] only focuses on two modalities and includes 6 XAI methods and 5 metrics. In addition, they do not address all the different evaluation pitfalls and metric disagreements as we do. Our selected scope includes all important design parameters affecting the evaluation, with the largest number of methods and metrics to date, while simultaneously focusing on non-overlapping computer vision modalities.
>
> In response to this feedback, we broadened the discussion section including implications for other modalities.
>
> **Answer to Cons 2:** Our study focuses on the widely used attribution and attention methods, which make up a substantial majority of the currently used feature importance methods in practice.
> By extensively focusing on these two method subgroups we cover all important methods for practitioners. In addition, they are well-supported by current evaluation metrics, whereas methods like symbolic metamodels remain niche with significant implementation challenges, especially for unconventional modalities such as volumes.
>
> We make these differences and complications now more explicit in the discussion of the paper.
>
> **Answer to Cons 3:** We focused on the three most informative and applicable evaluation criteria. The criteria of localization, which is closely connected to complexity, would be only applicable to image and volume datasets including bounding boxes as it would be infeasible for us to label all datasets ourselves. In addition, the criteria would only be very limited applicable to Point Clouds in general as there are no bounding boxes in the point cloud modality, and possible region segmentations over points do not cover the information on which region is most important for the classification task.
>
> We prioritized the three most informative and applicable evaluation criteria for our analysis. The criterion of localization, which is closely related to complexity, is specific to image and volume datasets that include bounding boxes. Labeling all datasets ourselves would be impractical, and moreover, this criterion is minimally applicable to point clouds, which lack bounding boxes and where regional segmentations fail to indicate the most significant regions for classification tasks.
>
> Additionally, axiomatic evaluation is unsuitable in contexts involving numerous XAI methods, as these methods do not uniformly adhere to the same axioms. In fact, only about half of the methods explicitly base their operations on axioms. Furthermore, while the confirmation of a method’s adherence to its foundational axioms is valuable for academic inquiry into mathematical behaviors, it offers limited practical guidance for practitioners selecting a method.
>
> In summary, our approach selectively incorporates the most informative criteria and deliberately excludes rather niche criteria that neither contribute significant new insights nor apply universally across all XAI methods and modalities. Based on the feedback, we extended our motivation of excluding these criteria in the discussion section of the manuscript.
>
> **Answer to Cons 4:** Thank you for pointing out this recently published study. This study utilizes four metrics focused on evaluating faithfulness and two metrics that require (pseudo) ground truth explanations. While informative, this study operates on a comparatively small scale and includes averaging the normalized scores of all metrics, a procedure that we argue in our answer to all reviewers above has specific shortcomings, especially with such a small set of metrics.
>
> Despite these differences, we have acknowledged the relevance of this work as it is the first work looking at more than one modality. We incorporated it into Table 1 of our manuscript and provided a comparative analysis of the results in Section 5, further illustrating the distinct approaches and outcomes between our study and the referenced work.
>
> References:\
> [NeurIPS, 2023] Li et al. (2023). M^4: A Unified XAI Benchmark for Faithfulness Evaluation of Feature Attribution Methods across Metrics, Modalities and Models.

---

> > ### Comment · Reviewer_U621 · 2024-05-11
> > **FDR control for a test**
> >
> > Thank you for your feedback! I found one more issue on the Levene tests of significance outlined on page 6. It is gratifying to see the structured setup for inference you've highlighted. However, regarding the addition of a significance test, your suggestion to develop methods for controlling the False Discovery Rate (FDR) is indeed insightful. Should we decide to compute p-values for a specific hypothesis, implementing an adjustment method such as the Benjamini-Hochberg procedure to refine p-value estimates would be a nice try in ensuring effective FDR control.

---

> > > ### Comment · Action_Editor_UVHU · 2024-06-06
> > > **Please read the rebuttal and make the official recommendation**
> > >
> > > Dear Reviewer U621,
> > >
> > > It has been two weeks late for you to submit the official recommendation. Please read the feedback from authors and make the official recommendation as soon as possible.
> > >
> > > Thank you.
> > >
> > > Action Editor

---

### Author Response · Authors · 2024-05-03
**Answer to all reviewers (1/2)**

Dear Reviewers,\
We sincerely thank all the reviewers for their insightful and comprehensive feedback. We appreciate the consensus among the reviewers regarding the significance of the benchmark and the importance of the results for the community. At the same time, concerns have been raised by reviewers “krZA” and “pcne” regarding the aggregation of metrics. In fact, we share these concerns and they even build the basis for our “metric analysis” (Section 3), where we identify and overcome crucial limitations of current aggregation schemes. The received feedback helped us to now communicate this nuanced perspective and motivation more clearly, and we explain this critical update here in detail.

**Improving the current state of Aggregation** \
We recognize there was a flaw in our communication regarding our contributions. The practice of averaging over standardized evaluation scores from a set of metrics is not a new proposal from our side, but an established approach in the XAI evaluation domain (see e.g. Li et al. [NeurIPS, 2023], Hedström et al. [JMLR, 2023] or Hesse et al. [IEEE/CVF, 2023]). In XAI, averaging different metrics is a means to approximate an underlying criterion for ensuring trustworthiness in the absence of ground truth information. This approach builds on the interpretation of various metrics as varying perspectives and mathematical interpretations of the same underlying criterion.

This current practice in the field has three major limitations, which we address in our work:

**Limitation 1:** Current practice aggregates only over small sets of metrics, which comes with an increased risk of biases due to a high dependence on metric selection and individual metric behavior, as especially the mean is not robust to outliers in such small samples.

We start addressing the first limitation by including the to-date largest scale of diverse and relevant metrics, minimizing the risk of selection bias and the influence of outliers. In fact, our study significantly expands the scope of analysis by including four times as many metrics and over twice as many XAI methods compared to the next most comprehensive study, which is further limited to only two modalities [NeurIPS, 2023].

It is important to note that we systematically chose the metrics we used; they were selected for their relevance to the modality, such as excluding metrics specific to natural language data. We also chose them based on three widely accepted criteria and their recognized importance in the community, as shown by frequent citations and inclusion in prominent libraries. As our motivations for metric selection are not specified in detail we added a more detailed explanation and motivation for each criterion and inclusion of metrics in Section 2.

**Limitation 2:** Aggregation of normalized scores can be flawed due to unbounded metrics, inconsistent interpretations, and sensitivity to outliers and distribution skewness.

Current studies perform aggregation over metrics by first normalizing or standardizing scores from each metric (e.g. z-score or min-max scaling), which is known to have several flaws: metrics can be unbounded, interpretations of measurements may differ across metrics even when scaled similarly, and the aggregation can be influenced by outliers and skewed distributions. To cite Colombo et al. [NeurIPS, 2022]: *“However, taking the mean [across scores values] is seriously flawed since the different metrics are usually not on the same scales and can even be unbounded. Even a pre-processing renormalization scheme would fail to capture the intrinsic difficulty of the tasks.”*.

We address this limitation by employing a “rank-then-aggregate” scheme, which is already well established for large-scale evaluation of model performance [Nature Com, 2018] [NeurIPS, 2022] [AAAI, 2021] [ICDM, 2005] as ranking statistics are more robust against skewed distributions and outliers compared to the raw scores. It can also be argued that rankings have higher interpretable value because they simplify comparison by clearly indicating the relative performance or position of methods, eliminating the need to understand varied scales or units (see Rosset et al. [ICDM, 2005]).

While the aggregation of ranks is not without flaws, it additionally possesses the beneficial property of highlighting strong trends across metrics. For a method to achieve either an average high or low rank, all included metrics must agree to a large extent in their evaluations. Given that rankings have inherent upper and lower bounds, a high degree of disagreement among them (indicated by a comparatively high standard deviation, SD) will naturally converge the average rank to a more intermediate (i.e. mean) rank. This mechanism minimizes the possibility of a "good but unstable" (reviewer “pcne”) rank. We leverage this behavior by focusing exclusively on strong trends where there is agreement across all metrics.

(1/2)

---

> ### Author Response · Authors · 2024-05-03
> **Answer to all reviewers (2/2)**
>
> **Limitation 3:** Current studies discard the standard deviation during aggregation, which contains crucial information about a method’s performance.
>
> Methods that receive intermediate average scores or ranks may attain such a categorization either because all metrics consistently agree or because they exhibit significant disagreement. The SD is necessary to resolve such ambiguities as it communicates the uncertainty behind an aggregation.
>
> Rank aggregation alone does not overcome the inherent limitations of mean aggregation, particularly its failure to distinguish whether an intermediate rank results from agreement or disagreement between metrics. We believe that understanding why metrics disagree and the situations in which this occurs is vital for evaluating XAI. To address this, we include the SD among metrics as an indicator of uncertainty for each mean aggregated rank. We conduct a detailed analysis for cases with high SD to identify which metrics disagree and explore the reasons behind these discrepancies.
>
> Responding to the feedback from reviewers, we have significantly expanded our metrics analysis and XAI benchmark to include reasons behind metric disagreements and the specifics of such discrepancies. It is important to note that a significant amount of our paper now focuses on analyzing metric behavior and enhancing the robustness of evaluations—areas that have received only sporadic attention in previous benchmarks and studies.
>
> The determination of a high SD is not trivial as reviewer “pcne” also remarked. To determine if a variance is large in general, we deployed the Levene Test to test if a variance between metric ranks is significantly smaller than the variance of a random ranking. However, the remaining focus of our work is less on whether an XAI method exhibits in general high or low SD between metric ranks, but more on how this SD compares with other methods under various conditions. Thus we use the quantiles of the SD distributions to determine threshold values within each evaluation criterion. We use this approach to outline high (i.e. >0.85 Quantile) and low SD (i.e. <0.15 Quantile) throughout the paper and color the SD accordingly in the newly added table in Table 2.
>
>
> **List of all improvements to the manuscript**
> - Reformulating metric analysis (section 2) to better emphasize motivation and problems with the current state of the art, as well as our solution.
> - A new subsection in section 2 about why metrics disagree including two new figures
> - Better emphasizing our motivation in the introduction (section 1)
> - Adding a second table complementing Table 2 showing the average standard deviation for each setting and highlighting differences in SDs by quantiles per criteria
> - Incorporating information about metric disagreement into the XAI benchmark (section 4) to highlight and understand uncertain ranking results, refining our interpretations accordingly.
> - Adding motivation for metric selection and the advantages of ranking statistics in section 2.
>
> We hope that these clarifications and enhancements address the concerns raised by the reviewers and demonstrate the thoroughness and rigor of our work. All raised weaknesses not addressed in this paragraph are addressed point-by-point below for each reviewer.
>
> We uploaded a preliminary revision of our manuscript along with this response. Changes to the manuscript are highlighted in orange. Based on the final outcome of the reviewer discussion, we will update and refine this revision.
>
> References:\
> [NeurIPS, 2023] Li et al. (2023). M^4: A Unified XAI Benchmark for Faithfulness Evaluation of Feature Attribution Methods across Metrics, Modalities and Models. \
> [JMLR, 2023] Hedström et al. (2023). An explainable ai toolkit for responsible evaluation of neural network explanations and beyond.\
> [IEEE/CVF, 2023] Hesse et al. (2023). FunnyBirds: A Synthetic Vision Dataset for a Part-Based Analysis of Explainable AI Methods.\
> [NeurIPS, 2022] Colombo et al. (2022). What are the best Systems? New Perspectives on NLP Benchmarking.\
> [Nature Commun, 2018] Maier-Hein et al. (2018) Why rankings of biomedical image analysis competitions should be interpreted with care.\
> [AAAI, 2021] Mishra et al. (2021). How Robust are Model Rankings: A Leaderboard Customization Approach for Equitable Evaluation.\
> [ICDM, 2005] Rosset et al. (2005). Ranking-based evaluation of regression models.
>
> (2/2)

---

### Author Response · Authors · 2024-05-08

Dear reviewers, many thanks again for taking the time to review our work. Since we received no further responses to our comments and the significant updates in the manuscript, we assume that all points of criticism have been addressed successfully. Kindly let us know if any issues remain.

---

> ### Comment · Reviewer_pcne · 2024-05-11
> **Further comments on the authors' response**
>
> Thank the authors for their response. I have carefully read the response. However, my main concern (i.e., the technical correctness of simply using mean aggregation to incorporate diverse metrics) is not addressed. The authors fail to directly demonstrate (discuss) the technical soundness, but try to do it by citing previous work and claiming that it is “an established approach” in XAI.
>
> Such response cannot convince me. Specifically,
>
> (1) An approach being adopted in previous work, doesn't necessarily mean it's correct.
>
> (2) I do not agree that the mean aggregation over different metrics is an “established” approach. In fact, in machine learning field, researchers usually use mean aggregation for a **single metric on different datasets**, instead of averaging over different metrics. Because it is generally believed that different metrics evaluate from different perspectives, and simply using mean aggregation has a high risk to result in wrong conclusions.

---

> ### Author Response · Authors · 2024-05-13
>
> Dear reviewer “pcne”,\
> Thank you for taking the time to review our responses. There seems to be a fundamental misunderstanding about our work, especially the difference between metric scores and metric ranks.
>
> Addressing (1):\
> We completely agree with this statement. However, while the statement implies that we agree with the current approach of metric aggregation, the opposite is the case: Importantly, we cite this current status quo of metric aggregation in XAI to highlight its flaws and contrast it against our approach aiming to overcome these flaws. We show in our manuscript, and the comments to the reviewers, how the current status quo of mean aggregation of *scores* is flawed (i.e. due to unbounded metrics, inconsistent interpretations, and sensitivity to outliers and distribution skewness). We then show that the mean aggregation of *ranks* together with the reporting of the resulting standard deviation between the metrics can overcome these flaws.
>
> Averaging ranks is technically not problematic as all rankings between metrics are on the same scale and have the same interpretation per value (i.e. the smallest rank is most favorable and the largest the least), which is not the case for the raw scores. Another important difference to common ML performance evaluation is the lack of ground truth in XAI. To compensate for this lack of ground truth, there are typically various metrics approximating the same task criterion (e.g faithfulness) with varying mathematical implementations. As a consequence, aggregating ranks over these metrics *within the same criterion* is conceptually more coherent than a rank aggregation over conceptually different classification metrics like accuracy versus F1-Score would be.
>
> Addressing (2):\
> We do not claim the approach of averaging metric scores is established in machine learning in general but in XAI evaluation as all three large-scale XAI evaluation benchmarks published during the last two years use this approach. As stated above, we agree that this approach is flawed for the XAI setting, which differs from the standard ML evaluation setting where typically scores from the same metric are averaged across different datasets.
>
> When, however, aggregating over various evaluation metrics, machine learning benchmarks and competitions often employ aggregation of ranks. See for example this in-depth analysis [1] or ranking strategies of popular competitions such as the BraTS Challenge [2] or the NeurIPS Multi-modality Cell Segmentation Challenge [3]. Thus, we stated rank aggregation as an established approach in large-scale model performance evaluation even though it is of course not applied in every benchmark and competition. While in these common ML benchmarks and competitions, aggregation over metrics is used to derive a robust final ranking for identifying the best model, we are the first to deploy this “rank-then-aggregate” scheme on XAI evaluation with the goal to distill generalizing trends from a large number of rankings.
>
> [1] Maier-Hein et al., Nature Communications (2018), Why rankings of biomedical image analysis competitions should be interpreted with care.\
> [2] Li et al. (2023), The Brain Tumor Segmentation (BraTS) Challenge 2023: Brain MR Image Synthesis for Tumor Segmentation (BraSyn).\
> [3] Ma et al. (2024), The Multi-modality Cell Segmentation Challenge: Towards Universal Solutions.

---

### Note · Authors · 2024-06-06

**Comment:**

Dear Editors and Reviewers,

Thank you very much again for your valuable time to review our work. Your thoughtful comments helped us to improve the manuscript significantly.

The initial mixed reviews and the lack of response to our rebuttal and the updated manuscript make us believe that the probability of acceptance at TMLR is low. As pointed out by the Action Editor, there has been a 2-week delay in the review process, which we did not anticipate in our publication timeline. Unfortunately, and as communicated last week, we cannot wait longer for the final decision. Therefore, we need to withdraw our work today from TMLR to comply with dual-submission standards.

**Withdrawal Confirmation:**

I have read and agree with the venue's withdrawal policy on behalf of myself and my co-authors.